# MAP4K4 regulates forces at cell–cell and cell–matrix adhesions to promote collective cell migration

Lara Elis Alberici Delsin[1], Cédric Plutoni[1], Anna Clouvel[2], Sarah Keil[1], Léa Marpeaux[1], Lina Elouassouli[1], Adele Khavari[2], Allen J Ehrlicher[2], Gregory Emery[1,3]

**Collective cell migration is not only important for development and tissue homeostasis but can also promote cancer metastasis. To migrate collectively, cells need to coordinate cellular extensions and retractions, adhesion sites dynamics, and forces generation and transmission. Nevertheless, the regulatory mechanisms coordinating these processes remain elusive. Using A431 carcinoma cells, we identify the kinase MAP4K4 as a central regulator of collective migration. We show that MAP4K4 inactivation blocks the migration of clusters, whereas its overexpression decreases cluster cohesion. MAP4K4 regulates protrusion and retraction dynamics, remodels the actomyosin cytoskeleton, and controls the stability of both cell–cell and cell–substrate adhesion. MAP4K4 promotes focal adhesion disassembly through the phosphorylation of the actin and plasma membrane crosslinker moesin but disassembles adherens junctions through a moesin-independent mechanism. By analyzing traction and intercellular forces, we found that MAP4K4 loss of function leads to a tensional disequilibrium throughout the cell cluster, increasing the traction forces and the tension loading at the cell–cell adhesions. Together, our results indicate that MAP4K4 activity is a key regulator of biomechanical forces at adhesion sites, promoting collective migration.**

## Introduction

Collective cell migration is a highly coordinated process important for development, tissue homeostasis, and wound healing. It can take several forms because cells can migrate as sheets, streams, or clusters (1, 2, 3). Collective cell migration can also occur under pathological conditions, as during cancer metastasis. Increasing evidence has demonstrated that cancer cells migrating collectively are more efficient at forming metastases compared with individualized cells, as cell clusters are better at invading tissues and

surviving in a new environment (4, 5, 6). Moreover, patients presenting circulating tumor cell clusters have worse survival rates (7).

To migrate collectively, cells interact with their environment, frequently the extracellular matrix, and their neighbor-migrating cells. To interact with their environment, cells form focal adhesions, large complexes of proteins that bridge the cell cytoskeleton to the extracellular matrix proteins, mainly through transmembrane proteins of the integrin family (8, 9). The contact with the neighbor cells is mediated by cell–cell junctions, which connect the actomyosin cytoskeleton between two or more cells through cadherins and catenins (2, 10).

Both focal adhesions and adherens junctions are mechanosensitive platforms, where cells can apply, sense, and transmit forces. Forces are generated by the activation of myosin II, which binds to actin filaments, promoting cytoskeleton contraction. These forces can be transmitted through mechanosensitive protein–protein interactions at focal adhesions to generate traction forces, or at adherens junctions to promote intercellular stresses (10, 11, 12).

During collective migration, the adhesion sites and the cytoskeleton structure are constantly remodeled, with a net force that causes the cells to move. To metastasize collectively, neighboring cancerous cells need to coordinate the dynamics of cell–cell adhesions, promoting detachment from the primary tumor whereas retaining cluster cohesion (6). Moreover, by remodeling their adhesions with the substrate and surrounding cells, cells can regulate the generation of traction forces and the transmission of stresses throughout the group (10). How this communication mechanism is regulated is not entirely understood, and the central proteins that coordinate this process are still need to be identified.

In this context, the mitogen-activated kinase kinase kinase kinase 4 (MAP4K4) influences collective cell migration in different model systems. MAP4K4 is a serine/threonine protein kinase of the Ste-20 family that has been involved in the regulation of several signaling pathways. Therefore, MAP4K4 deregulation associates to different pathologies, including cancer (13, 14). The *Drosophila* orthologue of *MAP4K4*, *misshapen*, was highlighted as key for the coordination of protrusion extension and rear retraction during

[1]Vesicular Trafficking and Cell Signalling Research Unit, Institute for Research in Immunology and Cancer (IRIC), Université de Montréal, Montréal, Canada   [2]Department of Bioengineering, McGill University, Montreal, Canada   [3]Department of Pathology and Cell Biology, Faculty of Medicine, Université de Montréal, Montréal, Canada

Correspondence: gregory.emery@umontreal.ca

border cells cluster migration (15). It was also shown to regulate focal adhesion dynamics in both *Drosophila* and mammalian cells, driving follicle epithelial cell migration during morphogenesis (16) and vascularization during mouse embryogenesis (17). Specifically, MAP4K4 induces integrin recycling, and different molecular models for this function have been proposed (17, 18, 19). Importantly, MAP4K4 is overexpressed in several solid tumors and frequently associated with a poor survival rate (13, 20, 21, 22). Increasing evidence places MAP4K4 as a pro-metastatic regulator, inducing cancer cell migration (13). However, the role of MAP4K4 in collective migration of cancer cells has not been addressed.

Here, we investigate the role of MAP4K4 in the regulation of the collective migration behavior of cancer cells, using the squamous epidermoid carcinoma cell line A431 as a model for cluster migration. We show that MAP4K4 is required for cluster migration through the regulation of protrusion and retraction dynamics. Specifically, MAP4K4 depletion stabilizes both the actomyosin cytoskeleton and focal adhesions, which results in higher traction forces at the substrate. We further report that higher forces are also applied on adherens junctions, and that intercellular stresses are regulated by MAP4K4. Interestingly, we found that MAP4K4 is localized at adherens junctions where it promotes junction disassembly when overexpressed and, consequently, reduces force transmission. Overall, our work shows that MAP4K4 coordinates the generation and transmission of forces during collective cell migration by regulating the stability of adhesion sites.

# Results

## MAP4K4 is required for the collective migration of carcinoma cells through protrusion and retraction dynamics

When cultured at a low density, A431 cells form clusters of 6–15 cells that migrate collectively as a cohesive entity (see the Materials and Methods section). Because of this property, A431 cell line was the primary model used for our study. To determine whether MAP4K4 is required for A431 cluster migration, we used CRISPR–Cas9 and two independent guides (sgRNA) to generate *MAP4K4* knocked-out cells (*MAP4K4* KO), or a non-target sgRNA sequence as control (sgNT) (Fig 1A). Migrating clusters grown on collagen–Matrigel were tracked over 5 h. *MAP4K4* KO reduced the instantaneous migration speed of clusters when compared with control sgNT (Figs 1B and S1A). Moreover, treating cells with GNE-495, a specific MAP4K4 kinase inhibitor (23), reduced the migration speed in a dose-dependent manner (Figs 1C and S1B and C), showing that the role of MAP4K4 in collective cell migration depends on its kinase activity.

To investigate how MAP4K4 regulates cell migration, we examined the actin cytoskeleton to understand cluster morphology. Control clusters presented both protruding and retracting cells at their periphery, characterized, respectively, by apparent F-actin "arches" at the protrusion base (arrows) or retraction fibers (arrowheads) (Fig 1D). On the other hand, both *MAP4K4* KO– and GNE-495–treated cells (from now on referenced as MAP4K4 loss of function—LOF) presented only cells with large, lamellipodia-like protrusions at the cluster periphery (Fig 1E and F). Consequently,

the morphology of the cluster was more circular (Fig 1G). Similar morphological changes of *MAP4K4* KO or GNE-495 treatment were observed when cells were treated with two other MAP4K4 inhibitors DMX-5804 (24) and PF-06260933 (25) (Fig S1D).

To gain insights into the processes regulated by MAP4K4, we tracked the margin of control and MAP4K4-inhibited cell clusters. We found that the displacement of the periphery was reduced after MAP4K4 inhibition (Fig S2A and B), meaning that protrusions and retractions events were less dynamic. Accordingly, the speed of both cellular extensions and retractions decreased (Figs 1H and I and S2C and D). This suggests that MAP4K4 regulates migration by promoting the dynamics of protrusion extensions and retractions across the cluster.

## MAP4K4 increases focal adhesion dynamics and regulates cytoskeleton organization

Previous works have shown that MAP4K4 regulates cell retraction by promoting focal adhesion disassembly in different cell types (17, 18, 19). To test if MAP4K4 is regulating focal adhesions disassembly in A431 carcinoma cells, we performed live imaging of the focal adhesion component paxillin fused to GFP. MAP4K4 inhibition increases stable focal adhesion and decreases assembly and disassembly rates, as shown by the highly stable GFP-enriched focal points over time (Fig S3A and B and Video 1).

Focal adhesions are formed as nascent adhesions at the front of the lamellipodium, and the subsequent recruitment of structural and signaling components induce their maturation while moving rearwards to the lamella (9). Mature focal adhesions bind to the cytoskeleton stress fibers through the actin-binding proteins α-actinin, zyxin, and VASP and generate traction forces at their distal tip (8, 26, 27). To further characterize the effect of MAP4K4 inhibition on focal adhesion, we stained A431 clusters for zyxin. MAP4K4 inhibition increased zyxin-positive focal adhesions, indicating that MAP4K4 regulates the dynamics of mature focal adhesions and prevents their accumulation (Fig 2A and B). Similar results were observed in cells KO for *MAP4K4* or using different MAP4K4 inhibitors (Fig S3C and D).

Because MAP4K4 LOF induces mature focal adhesion stabilization, and those are frequently bound to F-actin stress fibers, we questioned if MAP4K4 LOF would also affect the cytoskeleton organization in the cell protrusion or if the formation of stress fibers was defective in absence of MAP4K4 kinase activity. For that, we performed a detailed characterization of the cytoskeleton organization in MAP4K4 LOF clusters. Stress fibers are bundles of F-actin, which support mechanical tension, helping cells to contract and to regulate their adhesion to the substrate (28). They can be classified as ventral or dorsal and transversal arcs (Fig 2C) (28). Dorsal stress fibers extend from focal adhesions to the dorsal part of the cell and are enriched in α-actinin, which acts as a crosslinker to stabilize the actin filament bundles (29). Immunostaining of α-actinin and the focal adhesion protein vinculin showed that MAP4K4 LOF induces the accumulation and elongation of dorsal stress fibers originating from focal adhesions (Figs 2D–F and S4A).

Dorsal stress fibers support the highly contractile transversal arcs, coupling the actomyosin machinery to focal adhesions (30, 31). Transversal arcs are enriched in active myosin and undergo a

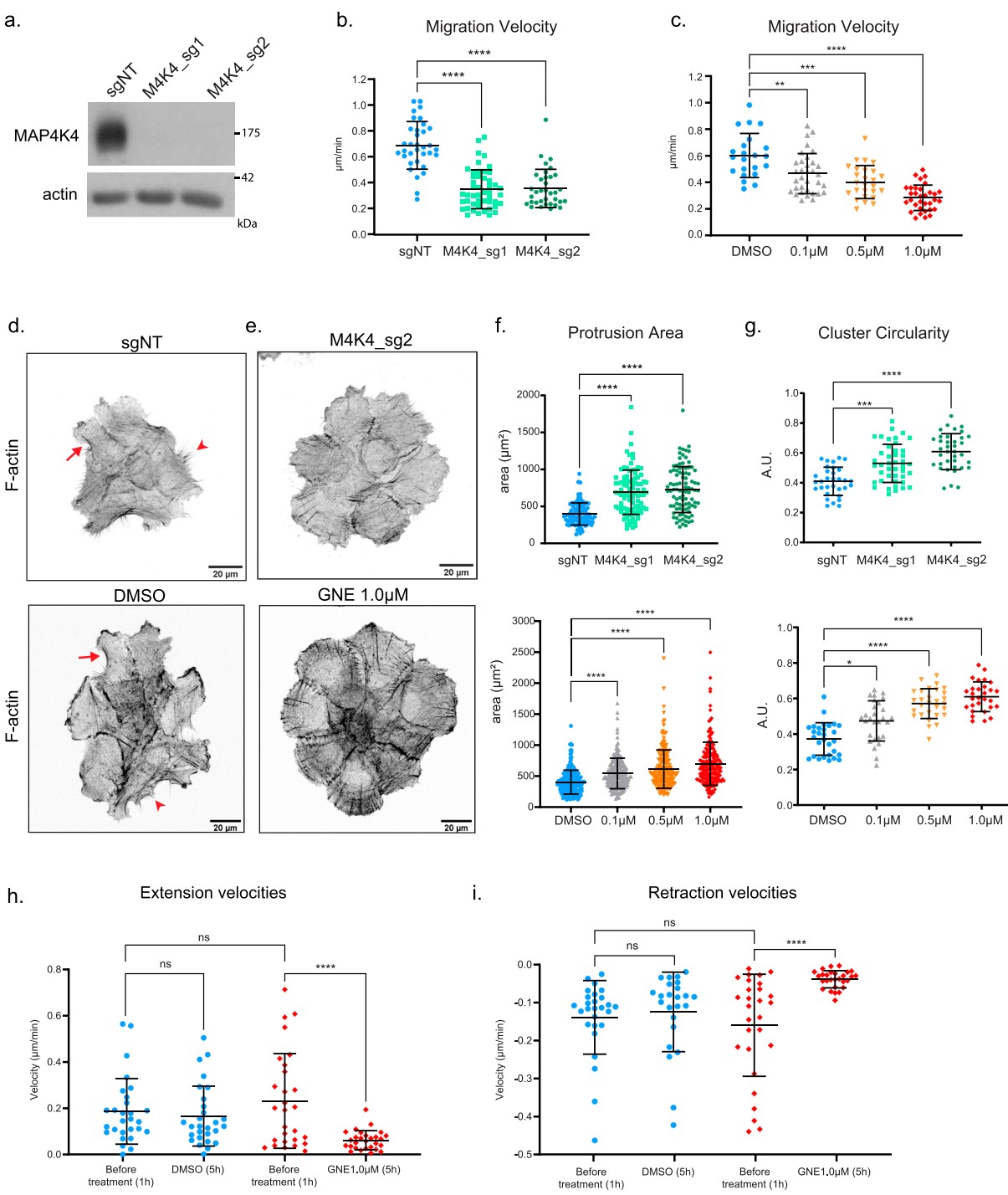

**Figure 1. CRISPR/Cas9– or GNE-495–mediated MAP4K4 inhibition impair A431 cluster migration through protrusion and retraction dynamics.**
**(A)** Representative immunoblotting of MAP4K4 and actin using lysates of A431 control cells (sgNT), or A431 cells KO for *MAP4K4* with two independent sgRNA (M4K4_sg1, M4K4_sg2). **(B)** Mean velocity of A431 clusters control or KO for *MAP4K4*, tracked over 5 h of migration. **(C)** Mean velocity of A431 clusters treated with DMSO or GNE-495 at different doses (0.1, 0.5, or 1.0 $\mu M$), over 5 h of treatment. Number of clusters analyzed (sgNT: 34, M4K4_sg1: 48, M4K4_sg2: 35, DMSO: 22, GNE 0.1 $\mu M$: 34, GNE 0.5 $\mu M$: 26, GNE 1.0 $\mu M$: 33), from three independent experiments. **(D, E)** z-scan projection of representative confocal images of F-actin stained A431 clusters, showing the differences in the actin cytoskeleton organization and in the morphology of clusters control (sgNT) or KO for *MAP4K4* (M4K4_sg2) or (E) clusters treated with DMSO or GNE-495 at 1.0 $\mu M$ for 24 h. Arrows represent the actin arches at protrusion bases and arrowheads indicate retraction fibers. **(F)** Protrusion area of control/*MAP4K4* KO cells or DMSO/GNE-495–treated cells with indicated doses. At least five clusters per experiment, three protrusions per cluster from three independent experiments were analyzed. **(G)** Circularity of control/*MAP4K4* KO cell clusters, or clusters treated with DMSO or GNE-495 at indicated doses. At least 25 clusters from three independent experiments were analyzed. **(H, I)** Mean velocity extension (H) or retraction (I) events at the periphery of the clusters before or after treatment with DMSO or GNE-495 at 1.0 $\mu M$, over 5 h of treatment. Number of clusters analyzed (DMSO: 28, GNE 0.1 $\mu M$: 26, GNE 0.5 $\mu M$: 26, GNE 1.0 $\mu M$: 28) from three independent experiments. All the data are presented as mean ± s.d. and tested by Kruskal–Wallis (*P < 0.05, **P < 0.01, ***P < 0.001, ****P < 0.0001).

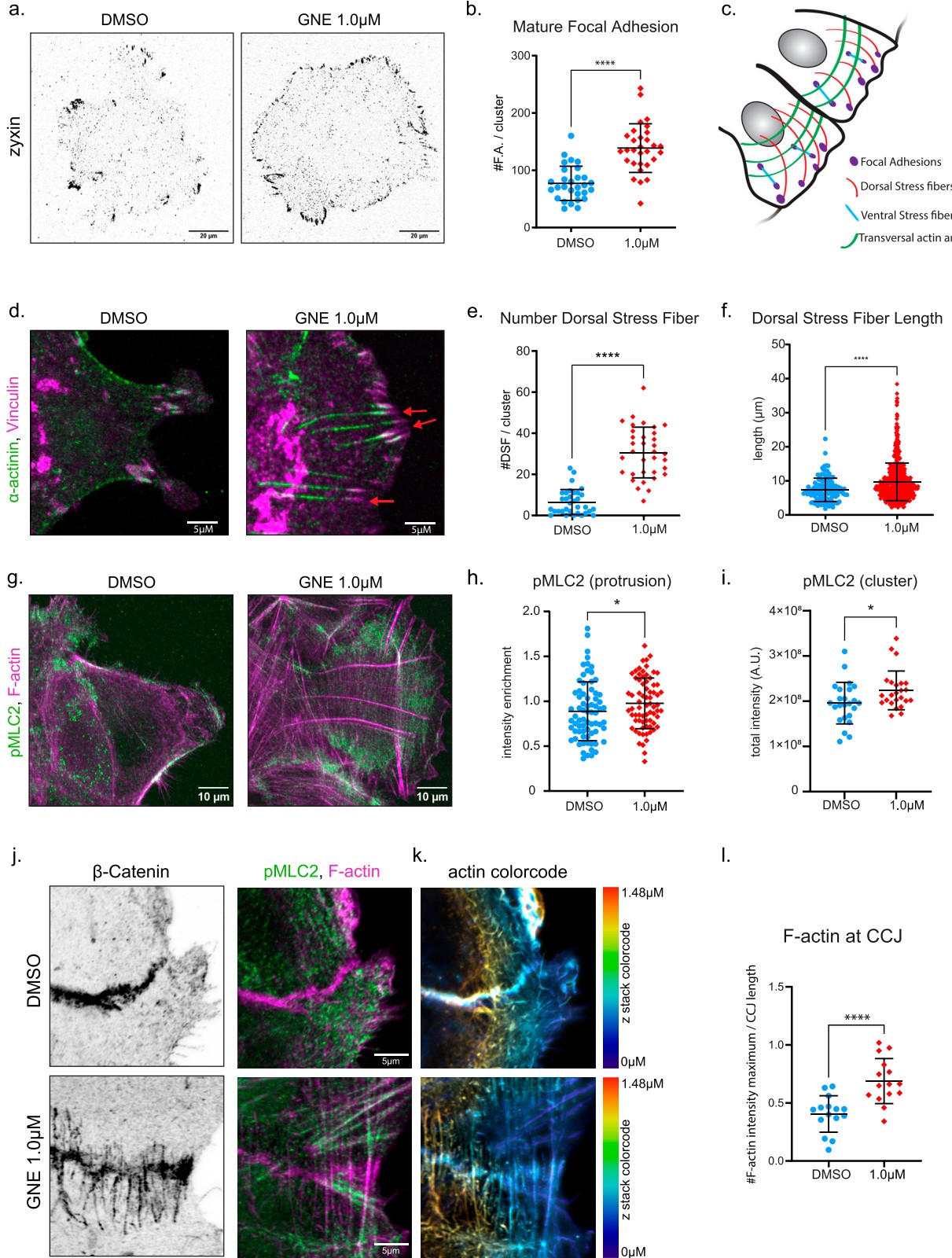

**Figure 2. MAP4K4 loss of function stabilizes focal adhesions and increases F-actin bundles.**
**(A)** Representative confocal images of A431 clusters treated with DMSO or GNE-495 at 1.0 $\mu$M stained for zyxin, a mature focal adhesion marker. **(B)** Number of zyxin-positive focal adhesions on clusters treated with DMSO or GNE-495 at 1.0 $\mu$M. At least eight clusters per experiment, from three independent experiments were analyzed. **(C)** Schematic representation of the different types of stress fibers. **(D)** Confocal z-scan projection of representative cell protrusion treated with DMSO or GNE-495 at 1.0 $\mu$M

retrograde flow movement at the protrusion lamella, exerting pulling forces at focal adhesions and ultimately enhancing traction forces at the cell–substrate (8, 31). Immunostaining of active myosin, using a specific antibody against phosphorylated myosin light chain 2 (MLC2), revealed that MAP4K4 LOF induces an accumulation of pMLC2 on transversal arches at the lamella (Fig 2G). Quantification of pMLC2 intensity shows an enrichment of active myosin at cell protrusion (Fig 2H). Extensive literature shows that myosin II activity is required at cell protrusions for focal adhesion maturation, stress fiber elongation, and increasing traction forces (8, 31, 32, 33). Therefore, this relocation of active myosin when MAP4K4 is inactive may, in turn, contribute to the maturation of the focal adhesion and elongation of stress fibers (8, 31, 34). Moreover, we observed a small, but significant increase in the total levels of active myosin II in MAP4K4 LOF (Figs 2I and S4B) despite Western blot analysis showed no difference in phosphorylation levels when relative to total myosin (Fig S4C and D).

Such alterations in the stress fibers organization indicate that MAP4K4 LOF not only promotes the stabilization of focal adhesion, preventing protrusions to retract, but this LOF also leads to a massive reorganization of the cytoskeleton at protrusions, increasing their stability, contractility levels, and potentially affecting the traction forces exerted on the substrate.

Interestingly, because of the collective properties of our chosen model of study, we were able to observe a reorganization of the cell cytoskeleton near the cell–cell junctions. Specifically, actomyosin fibers from the transversal arches are bound perpendicularly to the cell–cell junction, presenting a continuous organization between cells (Fig 2J). Those fibers accumulate all along the cell–cell junction, as shown by the z-color–coded image of F-actin staining (Fig 2K and L). Finally, a 3D analysis of F-actin distribution in A431 clusters shows that MAP4K4 inhibition induces the formation of a thick F-actin network on the dorsal region of the cluster. Surprisingly, MAP4K4 LOF has minimal effects on the formation of ventral stress fibers, which localize at cell–substrate interface and binds to focal adhesion on both fiber's edges (28) (Fig S4E and F). Therefore, our characterization shows that MAP4K4 LOF has a supracellular effect on the cluster actin organization, inducing the accumulation of different types of stress fibers and reorganizing their cell-to-cell connections. Interestingly, we show that this reorganization occurs almost exclusively through the dorsal region of the cells and is possibly complementary to the accumulation of focal adhesions.

## MAP4K4 decreases the generation of traction force during CCM

Because MAP4K4 LOF induced accumulation of mature focal adhesions, reorganized stress fibers, and enriched active myosin at cell protrusions, we investigated if the inhibition of MAP4K4 would impact the forces applied to the substrate, by using traction force microscopy (TFM) (35, 36). Treatment of A431 clusters with GNE-495 induced higher mean traction forces and mean strain energy when compared with treatment with DMSO (Fig 3A–E). Therefore, when MAP4K4 is inactive, cells exert higher traction forces on the substrate, showing that MAP4K4 activity releases tension at cell–substrate interface.

Moreover, we used our TFM data to estimate the intercellular stresses field in A431 clusters, by Bayesian inversion stress microscopy (BISM) analysis. Interestingly, we found that intercellular tensile stresses are substantially increased after inhibition of MAP4K4 (Fig 3F and G). Tissue level intercellular stresses are sensed and transmitted through adherens junction (37). Therefore, based on this increase of tensile stress, along with the reorganization of the actomyosin cytoskeleton at the cell–cell junction (Fig 2I–K), and the formation of a supracellular actin network (Fig S4A and B), we hypothesized that MAP4K4 may be necessary to regulate tension balance at cell–cell adhesions and to control forces transmitted among cells.

## MAP4K4 decreases tension at adherens junction

Aiming at understanding if MAP4K4 regulates tension at the cell–cell adhesions, we performed staining of the adherens junction marker p120-catenin. Morphological analysis revealed junction alterations in MAP4K4 LOF clusters. In control clusters, adherens junctions are mostly linear, whereas the junctions of MAP4K4 LOF cells present a tortuous morphology (Fig 4A–D).

A similar tortuous adherens junction shape was reported in endothelial cells. They can appear after chemically induced contractility and are perpendicularly bound to stress fibers of neighbor cells (38, 39), as seen in MAP4K4 LOF cells (Figs 2J, K, and 4B). Based on both, the increase on intercellular forces and the reshaping of adherens junction, we decided to further investigate whether MAP4K4 regulates tension loading on cell–cell adhesions.

To test that, we immunostained vinculin, a protein recruited to adherens junctions under tension, downstream of opening of the mechanosensitive protein α-catenin, which density

and stained for α-actinin (green) and vinculin (magenta). Arrows indicate F-actin fibers enriched in α-actinin that elongates from focal adhesions towards the dorsal part of the cluster, the so-called dorsal stress fibers. **(E)** Number of dorsal stress fibers per cluster treated with DMSO or GNE-495 at 1.0 μM. At least 30 clusters from three independent experiments were analyzed. **(F)** Length of dorsal stress fibers of clusters treated with DMSO or GNE-495 at 1.0 μM. All the dorsal stress fibers of at least eight clusters per experiment from three independent experiments were measured. **(G)** Confocal z-scan projection of representative cell protrusion treated with DMSO or GNE-495 at 1.0 μM and stained for F-actin (magenta) and pMLC2 (green), showing the differences in pMLC2 accumulation in DMSO or after GNE-495 treatment at 1.0 μM. **(H)** Ratio of the mean intensity of pMLC2 at protrusions over the mean intensity of pMLC2 at entire cluster. At least three protrusion per cluster, from eight clusters per experiments of three independent experiments were analyzed. **(I)** Quantification of total pMLC2 intensity in DMSO or after GNE-495 treatment. At least 24 clusters from three independent experiments were analyzed. **(J)** Confocal z-scan projection of representative cell–cell junction treated with DMSO or GNE-495 at 1.0 μM and stained for β-catenin, pMLC2, and F-actin, showing the perpendicular organization of F-actin relative to the junction orientation after GNE-495 treatment. pMLC2 is accumulated at the thick F-actin after GNE-495 treatment. **(K)** F-actin staining color coded by the position in the z-axis, showing accumulation of perpendicular thick F-actin along all the junction after GNE-495 treatment, including the more dorsal parts, indicated by the accumulation of filaments in yellow and red. **(L)** Relative abundance of perpendicular F-actin bundles along the cell–cell junction side of clusters treated with DMSO or GNE-495 at 1.0 μM, calculated as described in the Materials and Methods section. Around 15 cell–cell junctions from different cluster, from three independent experiments were analyzed. All the data are presented as mean ± s.d. and tested by Kruskal–Wallis (*P < 0.05, **P < 0.01, ***P < 0.001, ****P < 0.0001).

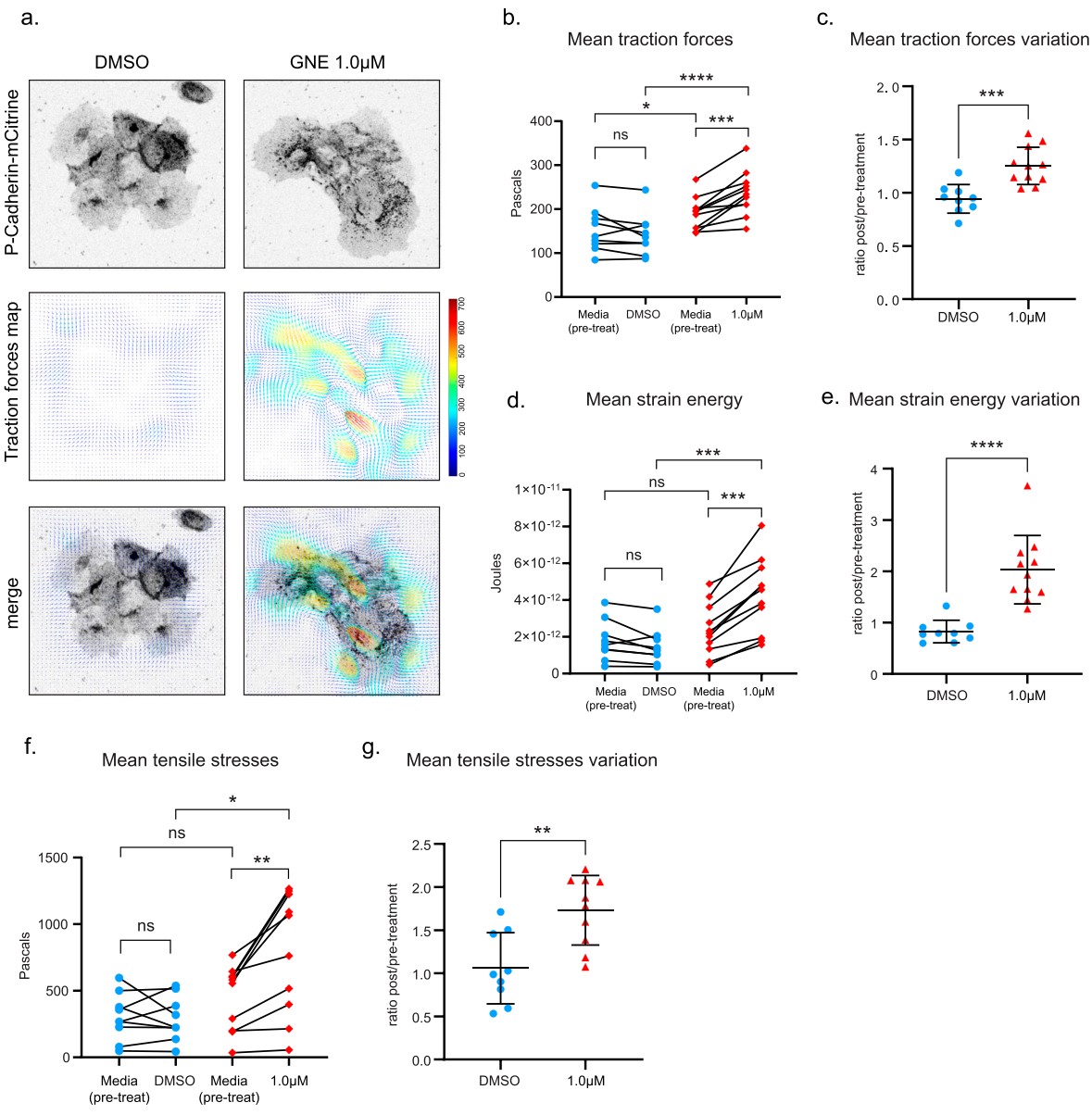

**Figure 3. MAP4K4 loss of function increases cluster traction forces and intercellular tensile stresses.**
**(A)** Confocal images acquired during time-lapse of A431 cells expressing P-cadherin–mCitrine, treated with DMSO or GNE-495 at 1.0 $\mu$M. Representative images of traction maps and overlay of traction maps and fluorescent images. **(B)** Mean traction forces of individual clusters before or after treatment with DMSO or GNE-495 at 1.0 $\mu$M (each data point is the average of three timepoints taken before treatment or between 4 h 30 m to 5 h of treatment). **(C)** Ratio of mean traction forces of individual clusters treated with DMSO or GNE-495 over the paired cluster before treatment. **(D)** Mean strain energy of individual clusters before or after treatment with DMSO or GNE-495 at 1.0 $\mu$M (each data point is the average of three timepoints taken before treatment or between 4 h 30 m to 5 h of treatment). **(E)** Ratio of mean strain energy of individual clusters treated with DMSO or GNE-495 over the paired cluster before treatment. At least nine clusters from three independent experiments were analyzed. **(F)** BISM analysis yielded the intercellular stresses, summarized here by tensile stresses. Mean tensile stresses were plotted as individual clusters before or after treatment with DMSO or GNE-495 at 1.0 $\mu$M (each data point is the average of three timepoints taken before treatment or between 4 h 30 m to 5 h of treatment). **(G)** Ratio of mean tensile stresses of individual clusters treated with DMSO or GNE-495 over the paired cluster before treatment. At least nine clusters from three independent experiments were analyzed. Data on (C, E, G) are represented as mean ± s.d. and tested by Mann–Whitney. Unpaired analysis on (B, D, F) was performed by Mann–Whitney test, whereas paired analysis was performed using Wilcoxon test (ns, nonsignificant; *P < 0.05, **P < 0.01, ***P < 0.001, ****P < 0.0001).

increases with tension (40, 41, 42). MAP4K4 LOF clusters accumulate vinculin at the adherens junctions compared with control (Fig 4E and F), suggesting that the adherens junctions are under higher tension when MAP4K4 is inactive. To understand if this phenotype is dependent on actomyosin contractility, we combine the GNE-495 treatment with a contractility inhibitor, the

Rho kinase inhibitor (Y-27632). The addition of Y-27632 abrogates the vinculin recruitment at adherens junction induced by MAP4K4 LOF, showing that this recruitment requires myosin-induced contractility (Fig 4G and H). Those results suggest that MAP4K4 decreases contractility and tension loading at the adherens junctions.

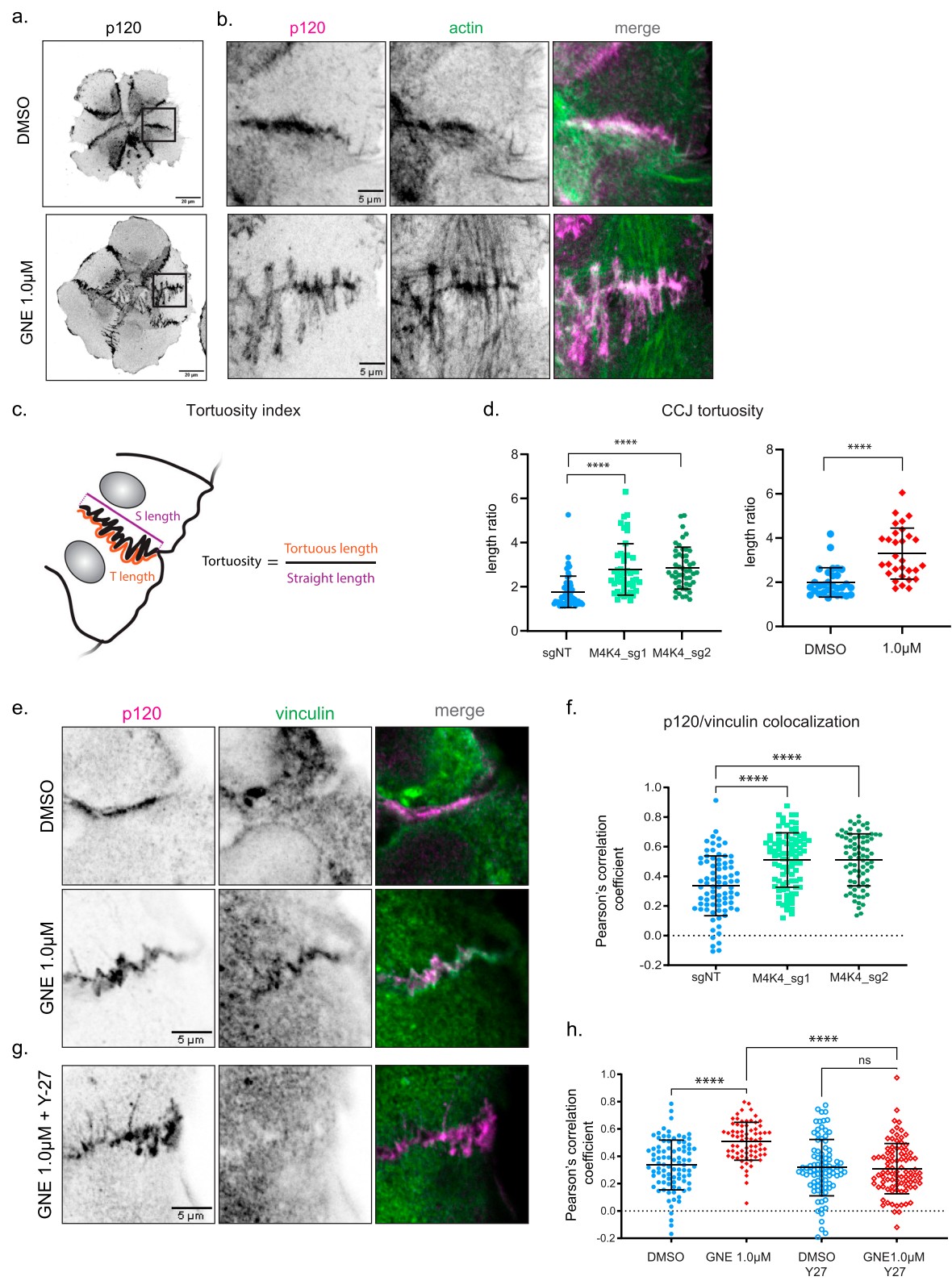

**Figure 4. MAP4K4 loss of function increases tension loading at cell–cell junctions.**
**(A)** z-scan projection of representative confocal images of p120-catenin stained A431 clusters, showing the differences in the cell–cell junction morphology of clusters treated with DMSO or GNE-495 at 1.0 μM. **(B)** Crops of the previous images, indicating parallel or perpendicular F-actin organization at the adherens junction. **(C, D)** Schematic representation of cell–cell junction tortuosity index calculation (D) Cell–cell junction tortuosity index for A431 clusters control (sgNT) or KO for *MAP4K4*, or treated with DMSO or GNE-495 at 1.0 μM. At least three junctions of five different clusters per experiment, from three independent experiments were analyzed.

Because contractile forces can affect junction remodeling and influence E-cadherin recruitment and turnover (43), we questioned if MAP4K4 inhibition would affect E-cadherin dynamics at adherens junctions. To understand this, we performed FRAP analysis on cells expressing E-cadherin–mRuby and compared recovery rates of control or GNE-495–treated clusters. Surprisingly, we did not observe a significant difference between the groups (Fig S6D), indicating that MAP4K4 inhibition does not affect E-cadherin dynamics.

### Loss of the MAP4K4 substrate moesin phenocopies MAP4K4 LOF at focal adhesions, but not at cell–cell junctions

In endothelial cells, MAP4K4 disassembles focal adhesions through the local phosphorylation of moesin (*MSN*) (17). Moesin is a member of the ezrin, radixin, moesin (ERM) family of proteins that link cortical actin to the plasma membrane (44). Phosphorylation of moesin by MAP4K4 ultimately induces integrin inactivation and focal adhesion disassembly (17). To test if a similar mechanism is at play in A431 cells, we first monitored phosphorylated ERM (pERM) using a phospho-specific antibody that recognizes a conserved phosphosite in all ERM proteins (45). MAP4K4 inhibition decreases pERM intensity at the focal plane containing the focal adhesions (Fig S5A and B). Concomitantly, we observed a reduction in the number and length of retraction fibers, which are ERM enriched structures that are formed when cells retract (Fig S5C and D). Surprisingly, pERM intensity measured at the adherens junction focal plane did not significantly change after MAP4K4 inhibition (Fig S5E and F).

If MAP4K4 acts exclusively through moesin, we would expect that *MSN* KO would phenocopy MAP4K4 LOF. To test this, we generated *MSN* KO cells by CRISPR–Cas9, using two independent guide sequences (sgRNA) (Fig S5G). The *MSN* KO cells presented an increase in zyxin-enriched mature focal adhesions (Fig S5H and I), suggesting that MAP4K4 phosphorylates moesin to regulate the dynamics of focal adhesion. However, *MSN* KO clusters do not present tortuous junctions (Fig S5J and K), showing that the loss of *MSN* is not sufficient to phenocopy the MAP4K4 LOF effect at adherens junction.

Altogether, our data suggest that MAP4K4 acts on a different substrate at adherens junction. Moreover, because *MSN* KO increases the number of mature focal adhesions but does not make cell–cell junctions more tortuous, we can hypothesize that the effect of MAP4K4 LOF at cell–cell junctions is not an indirect effect of focal adhesion stabilization. Hence, MAP4K4 might directly regulate forces at adherens junctions.

### MAP4K4 localizes at adherens junctions and regulates their disassembly

To examine if MAP4K4 has a direct effect on adherens junctions, we investigated its localization with an eGFP fusion to MAP4K4. We found that eGFP–MAP4K4 localizes at adherens junctions (Fig 5A), supporting our hypothesis that MAP4K4 directly regulates tension at junctions. In accordance with the literature (17), we also observed that MAP4K4 localizes at retraction fibers and cell rear in both single cells and cells in clusters (Fig S6A and B).

To explore how the localization of MAP4K4 at adherens junctions is regulated, we generated mutant constructs. MAP4K4 is composed of a kinase domain at its N-terminus, followed by a coiled-coil and an unstructured region, and a CNH domain (citron homology domain) at its C-terminal (14). We generated eGFP-tagged constructs with a kinase-inactive mutant (MAP4K4$^{D153N}$) and a C-terminal deletion of its CNH domain (MAP4K4$^{\Delta CNH}$) and explored their localization. We found that the recruitment of MAP4K4 at adherens junction is independent of its kinase activity; however, recruitment does require the CNH domain (Fig 5B and C). The localization of MAP4K4 and its mutants are similar in monolayers of MDCK cells, epithelial cells derived from canine kidney (Fig S6C), showing that the localization of MAP4K4 at adherens junction is not unique to a single cell type.

We tested the functionality of those constructs by performing rescue experiments in *MAP4K4* KO cells (Fig 5D). The WT form of MAP4K4 (MAP4K4$^{wt}$) completely restores the number of mature focal adhesion and junction linearity in KO clusters. However, KO cells expressing the kinase-inactive mutant still present a high number of mature focal adhesions and higher tortuosity rates. The expression of MAP4K4$^{\Delta CNH}$ does not rescue the accumulation of mature focal adhesions and induces only a partial rescue of junction tortuosity (Fig 5E and F). Those results indicate that both the CNH domain and the kinase activity are necessary for the full function of MAP4K4 at focal adhesions and adherens junctions. Moreover, the kinase-inactive construct corroborates that the effect observed upon GNE-495 treatment is specific to MAP4K4 kinase activity impairment.

To further characterize the role of MAP4K4 at cell–cell adhesion, we performed confocal time-lapse imaging of cells expressing both eGFP–MAP4K4 and E-cadherin–mRuby. Interestingly, we observed that MAP4K4 localizes at the cell–cell interface of detaching cells (Fig 6A and Video 2 and Video 3). To better understand this cell behavior, we imaged A431 cluster control or expressing eGFP–MAP4K4 during 2 h with a 1-min time resolution. Cells expressing eGFP–MAP4K4 were frequently found as single cells, and the percentage of cells detaching from clusters was significantly higher under this condition (Fig 6B and C and Video 4, Video 5, and Video 6).

---

**(E)** Representative confocal images of cell–cell junctions, showing vinculin accumulation at junctions of clusters treated with DMSO or GNE-495 at 1.0 $\mu$M. **(F)** Colocalization between p120 and vinculin intensities for clusters control (sgNT) or KO for *MAP4K4*, calculated by the Pearson's correlation coefficient. At least three junctions per cluster, from at least eight clusters per experiment, from three independent experiments were analyzed. **(G)** Representative confocal images of cell–cell junctions, showing loss of vinculin accumulation at junctions when GNE-495–treated clusters for 24 h were exposed to the ROCK inhibitor Y-27632 during 15 min. **(H)** Colocalization between p120 and vinculin in clusters treated with DMSO or GNE-495 at 1.0 $\mu$M alone or in combination with Y-27632 (2.5 $\mu$M), calculated by the Pearson's correlation coefficient. At least three junctions per cluster, from at least eight cluster per experiment, from three independent experiments were analyzed. Data on (D) are represented as mean ± s.d. and tested by Mann–Whitney test. Data on (F, H) are represented as mean ± s.d. and tested by Kruskal–Wallis test. (ns, nonsignificant; *$P < 0.05$, **$P < 0.01$, ***$P < 0.001$, ****$P < 0.0001$).

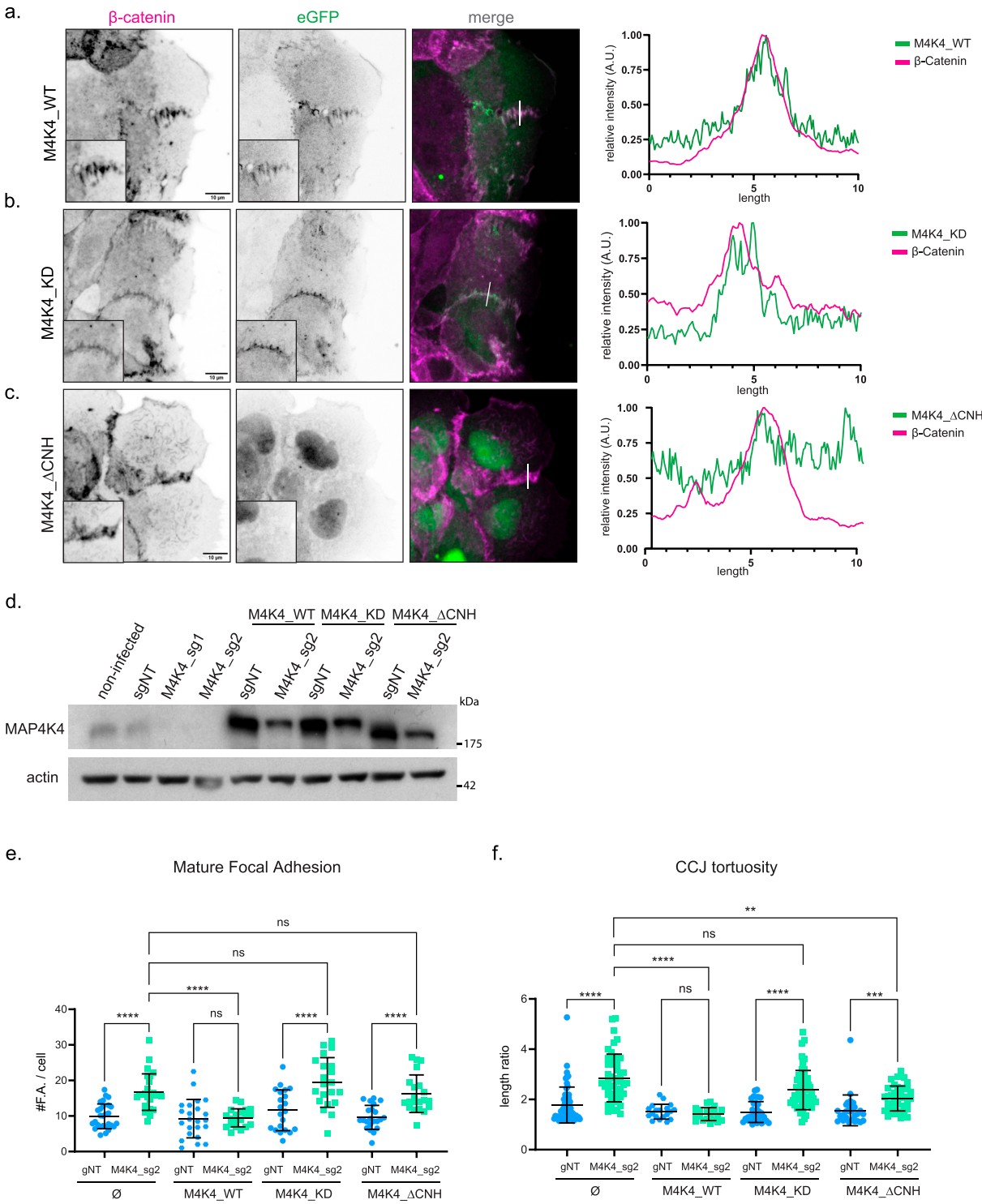

**Figure 5. MAP4K4 localizes at cell–cell junction, in a CNH-dependent manner, and both CNH domain and kinase activity are necessary for MAP4K4 role.**
**(A, B, C)** z-scan projection of representative confocal images of β-catenin stained A431 clusters, stably expressing (A) eGFP–MAP4K4_WT, (B) eGFP–MAP4K4 kinase dead (MAP4K4$^{D153N}$), or (C) deleted for the CNH domain (MAP4K4$^{ΔCNH}$). Line scan indicates the colocalization between β-catenin and MAP4K4. **(D)** Immunoblotting of MAP4K4 or actin for lysates of A431 cells controls (non-infected or sgNT), KO for *MAP4K4* (sg_1 or sg_2) alone or expressing eGFP–MAP4K4 WT, KD, or ΔCNH, resistant to sg_2. **(E)** Number of zyxin-positive focal adhesions for A431 clusters control (sgNT) or KO for *MAP4K4* (M4K4_sg2) and stably expressing eGFP–MAP4K4 WT, KD, or ΔCNH, resistant to sg_2. **(F)** Cell–cell junction tortuosity index for A431 clusters control (sgNT) or KO for *MAP4K4* (M4K4_sg2), and stably expressing eGFP–MAP4K4 WT, KD, or ΔCNH, resistant to sg_2. Data on (E, F) are represented as mean ± s.d. and tested by Kruskal–Wallis (ns, nonsignificant; *P < 0.05, **P < 0.01, ***P < 0.001, ****P < 0.0001).

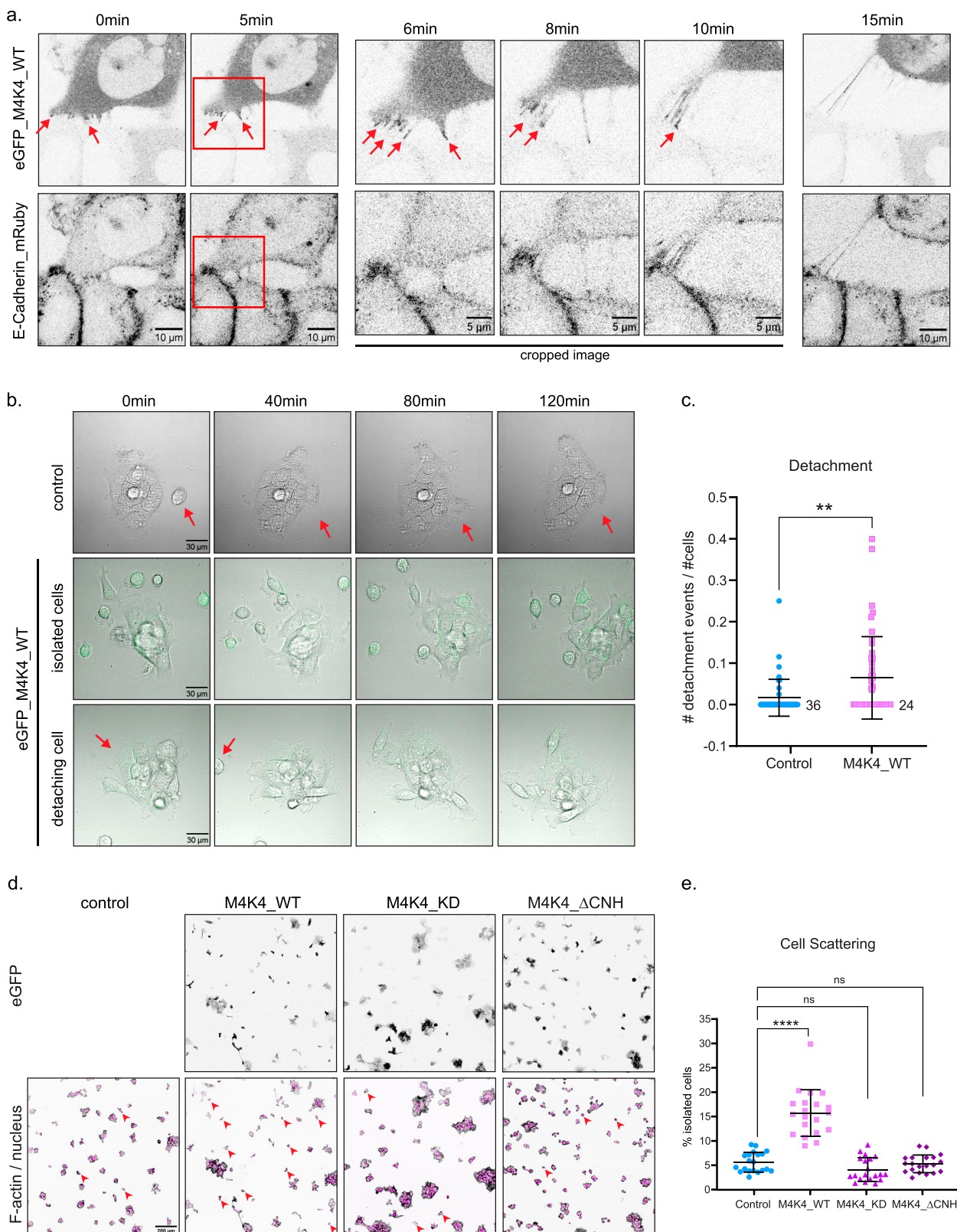

**Figure 6.   MAP4K4 accumulates in the interface of detaching cells and induces cell scattering.**
**(A)** Confocal images acquired during time-lapse of A431 cells expressing eGFP–MAP4K4 WT and E-cadherin–mRuby. Arrow indicates accumulation of eGFP–MAP4K4 at cell–cell contacts during cell detachment. Red squares indicate region that was cropped for timepoints 12, 16, and 20 min. Images were acquired every 30 s, during 30 min.
**(B)** Confocal images of DIC and eGFP, acquired during time-lapse of A431 control cells, or A431 cells expressing eGFP–MAP4K4 WT. Top panel represent control cells, middle

Therefore, MAP4K4 overexpression induces cell–cell adhesion disassembly and cell detachment.

We next investigated if MAP4K4 overexpression impacts on the cadherin turnover. For that, we performed FRAP analysis on cells expressing both eGFP–MAP4K4 and E-cadherin–mRuby. Because MAP4K4 overexpression increases detachment rates, we expected to see an increase in the cadherin turnover rates. However, we were unable to analyze cells with high expression levels of MAP4K4 as their cell–cell junctions were too dynamic or the cells were isolated. Our analysis was thus limited to low expressing cells with stable junctions. In these cells, we observed similar recovery rates for E-cadherin in control and MAP4K4 overexpressing cells (Fig S6E).

To determine if the effect of MAP4K4 on cell detachment was dependent on its kinase activity or on the presence of the CNH domain, we overexpressed the different MAP4K4 constructs presented in Fig 5 and analyzed the number of individualized cells among A431 clusters. Interestingly, we observed that MAP4K4$^{wt}$ overexpression increases the percentage of individualized cells when compared with control condition. Those single cells usually expressed high levels of MAP4K4 as observed by their GFP signal (Fig 6D). No difference in the number of single cells was observed when we overexpressed the kinase-inactive mutant or MAP4K4$^{\Delta CNH}$. Cells expressing high levels of these mutants, as detected by GFP, were found in cluster (Fig 6D and E). Together with its localization at disassembling cell–cell junctions and the increased rate of detachment events, our results suggest that MAP4K4 regulates cell–cell junction disassembly and promotes cell scattering.

# Discussion

In this work, we used the epidermoid carcinoma cell line A431 as a model to study the role of the kinase MAP4K4 in regulating the collective migration of cancer cell clusters in vitro. We show that MAP4K4 is a central regulator of force generation and transmission during collective migration, acting specifically on the disassembly of cell–substrate and cell–cell adhesions. Previous work showed that MAP4K4 regulates the disassembly and recycling of focal adhesions in individual cells, through different molecular mechanisms (17, 18, 19). Here, we found that MAP4K4, in the collective context, also induces focal adhesion turnover predominantly through moesin phosphorylation in A431 carcinoma cells. Moreover, we extend our analysis and show that when MAP4K4 is impaired, there are more mature focal adhesions, in parallel to an increase in the number of stress fibers at cell protrusions, and a decrease in protrusion dynamics. Therefore, MAP4K4 inhibition not only prevents cells to retract but also reorganizes the cytoskeleton network inside protrusions. Furthermore, we reveal that this process relocates active myosin to those protrusions, locally increasing contractility and inducing

cells to exert higher traction forces on the substrate (see the Graphical Abstract). Moreover, we observed emergent properties of MAP4K4 due to the collective context. The presence of protrusions around the entire cluster, as observed in MAP4K4 LOF, impairs the cell–cell coordination mechanism and blocks migration (46). Furthermore, MAP4K4 LOF induced a continuous F-actin organization through cell–cell junctions, favoring the formation of a dorsal F-actin network that may influence cell–cell communication. Therefore, in addition to stabilizing focal adhesions, MAP4K4 LOF promotes a cascade of cellular processes that culminates in the impairment of collective cell movement.

We also report for the first time that MAP4K4 can be recruited to cell–cell adhesions, and this seems to decrease their stability, tension loading, and transmission of forces. Previous works have suggested a role for MAP4K4 in cell–cell adhesion stability by studying endothelial cells' permeability. In that context, MAP4K4 depletion increases the resistance of the endothelial barrier (47, 48), which can indicate a tightening of cell–cell junctions. Here, we expand the understanding of the role of MAP4K4 at cell–cell junctions, showing that MAP4K4 LOF induces higher cell–cell junction tortuosity, which physically increases the adhesion area between the cells. We also characterize the recruitment of vinculin to cell–cell junctions in MAP4K4 LOF cells. Vinculin is recruited to the adherens junctions to help stabilizing the catenin–actin complex when strong pulling forces are exerted (40, 42, 49). Therefore, accumulation of vinculin in MAP4K4 LOF indicates higher junction stabilization and increase of tension loading, as validated by our intercellular stress calculations. Conversely, we show that MAP4K4 overexpression induces adhesion disassembly and cell scattering. Those results bring new insights about the role of MAP4K4 in balancing the adhesiveness and stability of adherens junctions under increased pulling forces.

It was reported that tension loading can either stabilize (50, 51) or reduce E-cadherin dynamics at cell–cell junction (52), depending on the model and the mechanism analyzed (43, 53). Despite the predicted link between adhesion stability and cadherin turnover rates, we could not observe significant difference on cadherin recovery by FRAP analysis in MAP4K4 LOF cells. We noticed that high levels of MAP4K4 expression promotes very dynamic cell–cell junction, making it technically challenging to photobleach, whereas lower expression levels presented no difference in recovery levels when compared with control. Therefore, we believe that MAP4K4 effect on adhesion stability or cadherin turnover rates depend largely to its level of expression, and the ideal condition to evaluate cadherin turnover is challenging to capture. Moreover, for a more comprehensive understanding of the molecular mechanism regulated by MAP4K4 at cell–cell adhesions, the recovery rates of several proteins in the junctional complex should be analyzed because their dynamics may vary independently, especially under different tension loadings and actin binding statuses (42).

panel represents MAP4K4-expressing cells that are isolated and very motile, bottom panel represents MAP4K4-expressing cells detaching from a cluster. **(C)** Mean number of cell detachment events per number of cells in the field of view, during 2 h time-lapse acquisition, every 1 min, on control cells or cells expressing eGFP–MAP4K4 WT. **(D)** Representative confocal images of A431 cells and clusters stably expressing eGFP–MAP4K4 WT, KD, or ΔCNH. Arrowheads indicate examples of isolated cells under each condition. **(E)** Percentage of cell scattering in A431 control, or overexpressing eGFP–MAP4K4 WT, KD, or ΔCNH. Data on (C, E) are represented as mean ± s.d. and tested by Kruskal–Wallis test (ns, nonsignificant; *$P$ < 0.05, **$P$ < 0.01, ***$P$ < 0.001, ****$P$ < 0.0001).

Interestingly, Pannekoek and colleagues reported that MAP4K4 acts downstream the small GTPase Rap2, which binds to the CNH domain of MAP4K4 (47, 54). Here, we describe that the recruitment of MAP4K4 to adherens junction depends on its CNH domain. Therefore, it is appealing to hypothesize that MAP4K4 may be recruited to adherens junctions by Rap2 to decrease its stabilization upon mechanical stresses. Accordingly, Meng and collaborators found that MAP4K4 acts downstream of Rap2 activation in response to substrate stiffness and focal adhesion stabilization (55). Therefore, it would be interesting to test if a similar mechanoresponse involving Rap2 and MAP4K4 is at play at adherens junctions.

Moreover, other proteins that contain a CNH domain are known to bind active Rho GTPases, as RhoA, Rac1, and Cdc42 (56, 57). Rho GTPases are master regulators of cell cytoskeleton and contractility (58, 59) and junction assembly and maintenance (58, 60, 61). Therefore, MAP4K4 might also be recruited to adherens junctions through the binding of Rho GTPases.

Despite the importance of MAP4K4 for cancer progression, little is known about its downstream direct targets. At focal adhesions, MAP4K4 seems to act predominantly through moesin phosphorylation. However, our data suggest that this is not the case at adherens junctions. Interestingly, in the endothelial leakage context, the phosphorylation of moesin at focal adhesions by MAP4K4 contributes to prevent gaps at cell–cell adhesions induced by inflammation, showing that the crosstalk between the two adhesive structures is important for endothelial cellular response. On the other hand, *MSN* depletion increased junctional stability in a MAP4K4-independent way, having an additive effect in MAP4K4 LOF cells (48). Our work brings further evidence that MAP4K4 acts at cell–cell adhesion independently from moesin because ERM phosphorylation levels was not affected at cell adhesion on MAP4K4 LOF, and *MSN* depletion did not affect adherens junctions morphology.

There are few known direct substrates that could be at play at adherens junction. However, none seem to explain MAP4K4 LOF phenotype in A431 clusters. One of the MAP4K4 targets is the actin nucleator Arp2 (62). However, it is unlikely that Arp2 is downstream of MAP4K4 at junctions because Arp2 is required for the maintenance of cell–cell junction and does not promote adhesion disassembly (63, 64, 65). Another known MAP4K4 substrate is LATS1/2, which regulates the mechanosensitive Hippo-signaling pathway. Although LATS is an interesting potential target of MAP4K4 in A431 cells, Meng and collaborators (66) showed that depletion of MAP4K4 alone is not sufficient to inactivate the Hippo transcriptional factor YAP. Therefore, it is unlikely that MAP4K4 regulates cell–cell adhesion through LATS, although some contribution of the Hippo pathway cannot be excluded.

Therefore, we think that MAP4K4 is acting directly at adherens junctions through a yet unidentified substrate. By exploring the Human Cell Map database (67), we found that MAP4K4 is predicted to localize at cell–cell junctions, consistent with our findings. Furthermore, MAP4K4 potentially interacts with several junctional proteins, including afadin, $\alpha$-catenin, and occludin. Future work will have to establish if MAP4K4 interacts physically or directly phosphorylates these proteins.

In conclusion, our findings show that MAP4K4 promotes the disassembly of both focal adhesions and adherens junctions. Focal adhesions and adherens junctions are indirectly connected through the cell cytoskeleton and several proteins are shared between the two structures (68, 69). The mechanical crosstalk between them has been explored (11, 70) and they can present cooperating (71) or antagonistic (72) responses. Here, we report a functional mechanism where MAP4K4 activity is central for balancing traction force generation through disassembly of focal adhesions, whereas it is also required to decrease the intercellular stresses and tension loading at the adherens junctions. Our work indicates that MAP4K4 is a key regulator of force balance to promote the collective migration of carcinoma cells. By regulating MAP4K4 expression and/or activation levels, cancer cell clusters can modulate their level of cohesion and thus the nature of their collective migration properties. Therefore, our results highlight a potential explanation for MAP4K4 pro-metastatic behavior.

# Materials and Methods

## Antibodies

For immunoblotting, the following primary antibodies were used: rabbit polyclonal anti-HGK (MAP4K4) at 1:1,000 (#3485; Cell Signaling Technology), rabbit polyclonal anti-phospho-myosin light chain 2 (Ser19) at 1:1,000 (#3671; Cell Signaling Technology), rabbit polyclonal anti-non-muscle myosin HC II-A at 1:1,000 (#909802; BioLegend), mouse monoclonal anti-actin at 1:10,000 (MAB1501; Millipore [C4]), rabbit polyclonal anti-moesin at 1:1,000 (#3150; Cell Signaling Technology [Q480]).

The following secondary antibodies were used: AffiniPure goat anti-rabbit (H+L) and goat anti-mouse (H+L) (111-035-144 and 115-035-062, respectively, used at 1/10,000; Jackson Immunoresearch).

For immunofluorescence, the following primary antibodies were used: rabbit polyclonal anti-phospho–ezrin (Thr567)/radixin (Thr564)/moesin (Thr558) at 1:250 (#3141; Cell Signaling Technology), rabbit polyclonal anti-phospho–myosin light chain 2 (Ser19) at 1:500 (#3671; Cell Signaling Technology), rabbit polyclonal anti-$\alpha$-actinin at 1:400 (A0761-2F; USBiological), mouse monoclonal anti-zyxin at 1:200 (sc-293448; SantaCruz [2D1]), rabbit polyclonal anti–p120-catenin at 1:200 (sc-13957; SantaCruz [H90]), mouse monoclonal anti-$\beta$-catenin at 1:100 (610153; BD Biosciences), mouse monoclonal anti-vinculin at 1:200 (V9131; Sigma-Aldrich).

The following secondary antibodies, at 1:1,000 dilution, were used: Alexa Fluor 488 goat anti-mouse IgG (A11029; Invitrogen), Alexa Fluor 488 goat anti-rabbit IgG (A11008; Invitrogen), Alexa Fluor 555 anti-mouse IgG (4499S; Cell Signaling Technology), Alexa Fluor 555 anti-rabbit IgG (4413S; Cell Signaling Technology). To stain nuclei, we used DAPI at 1 $\mu$g/ml (D8417-10 MG; Sigma-Aldrich).

To stain F-actin, we used Alexa Fluor 488 phalloidin at 1:1,000 (A12379; Invitrogen), Alexa Fluor 555 phalloidin at 1:750 (A34055; Invitrogen), Alexa Fluor 647 phalloidin at 1:50 (A22287; Invitrogen).

## Cell culture and lentivirus production

A431 (CRL-1555; ATCC), MDCK (CCL-34; ATCC), or HEK-293-T (CRL-3216) cells were cultured in DMEM (Sigma-Aldrich), supplemented with

10% FBS (Gibco) and 1% penicillin–streptomycin (Sigma-Aldrich) in 5% $CO_2$ air-humidified atmosphere at 37°C. For A431 cells, cluster confluence is reached by plating 1,500–2,000 cells/cm² and keeping cells in culture for 3 d. Cells were sporadically tested for mycoplasma.

The plasmids of interest were co-transfected with pCMV–VSVG and psPAX2 in HEK-293T to generate lentiviruses, using PEI. A431 or MDCK cells were infected with the virus particles using polybrene and selected using puromycin (0.5 μg/ml) or blasticidin (1 μg/ml).

## Constructs and CRISPR

pGIPz–HA–MAP4K4 construct was kindly shared by Vitorino and collaborators (17). The MAP4K4 sequence was cloned into a pLVpuro–CMV–N-EGFP (#122848; Addgene) lentiviral plasmid using the gateway system. The following primers were used to mutagenize MAP4K4 to kinase dead (D153N) (forward: 5′-GTGATTCACCGGAA-CATCAAGGGCC-3′, reverse: 5′-ATGATGAATGTGAAGATGTGCCAGTCCCC-3′) and to delete the CNH domain (forward: 5′-GGTGGCAGCAGTCAGGTT-TATTTCATGACCTTAGGCAGG-3′, reverse: 5′-TTTACGAATCTCCGGGGTGT-CACTCTGTGGCCTAGT-3′). Those constructs were used to produce lentivirus, as described before, and infected in A431 or MDCK to understand MAP4K4 localization and to perform overexpression experiments.

pLenti.PGK.Lifeact-GFP.W construct (#51010; Addgene) was used to produce lentivirus as described before. A431 cells were infected to stably express the Lifeact-GFP marker. Those cells were used for the migration assay and morphodynamical analysis (Fig 1).

Paxillin–pEGFP (#15233; Addgene) construct was cloned into a pLVpuro–CMV–N-EGFP (#122848; Addgene) lentivirus plasmid using the gateway system (ATTB sites were added at the paxillin–peGFP plasmid using the primers: forward: 5′-GGGGACAAGTTTGTA-CAAAAAAGCAGGCTTCGACGACCTCGACGCCCTGCTG-3′ and reverse: 5′-GGGGACCACTTTGTACAAGAAAGCTGGGTCCTAGCAGAAGAGCTTGAG-GAAGC-3′). This construct was used to produce lentivirus, as described before, and infected in A431 to measure focal adhesion dynamics.

Lentiviral expression plasmids for E-cadherin–mRuby3 and P-cadherin–mCitrine were kindly shared by Arnold Hayer (McGill University, Montreal). Lentivirus particles were produced by co-transfection of pCMV–VSVG, pMDLg, and pRSV-rev into HEK-293T cells and A431 cells were infected and selected with puromycin (0.5 μg/ml) to stably express the cadherins as cell–cell junction markers. Those cells were used for the traction forces microscopy assay and MAP4K4 localization time-lapse imaging.

MAP4K4 KO cells were generated using the pLenti.Cas9-blast (#52962; Addgene) construct and the following sgRNA constructs: MAP4K4_sg1 (#76263; Addgene) and MAP4K4_sg2 (#76264; Addgene). Control cells were generated using pLenti.Cas9-blast (#52962; Addgene) and the non-targeting control gRNA (#80263; Addgene). Lentivirus was produced as described before. Cells were co-infected with the Cas9 and sgRNA lentiviruses and co-selected with puromycin (0.5 μg/ml) and blasticidin (1 μg/ml).

pLVpuro–CMV–N-EGFP–MAP4K4 resistant to the MAP4K4_sg2 sequence was generated by introducing silent mutations into the targeted sequence. The following primers were used: forward: 5′-GTCAGCGCTCAGCTGGACAGGACTGTG-3′, reverse: 5′-

GCCGAAATCCACAAGTTTCACCTCTGCATTCTCAGTC-3′. Lentivirus was produced as described before. Cells were used for rescuing MAP4K4 LOF experiments.

Plasmids containing the sgRNA for *MSN*, *EZR*, and *RDX* and the control Rosa were a gift from Sebastien Carréno (IRIC, Montréal). Those plasmids also contain the sequence for Cas9. Cells were infected and selected using puromycin (0.5 μg/ml).

## Drug treatment

GNE-495 (HY-100343; MedChemExpress) was dissolved in DMSO and diluted in complete medium. Cells were treated at the doses 0.1, 0.5, or 1.0 μM for 24 h. The inhibitors PF-06260933 (HY-19562; Med-ChemExpress) and DMX-5804 (HY-111754; MedChemExpress) were also dissolved in DMSO and used at the doses 0.5 or 1.0 μM for 24 h. The Rho kinase inhibitor (Y-27632; Cell Signaling) was used to inhibit contractility after 24 h of GNE-495 treatment, at 2.5 μM for 15 min.

## Migration assay, live imaging, and morphodynamic analysis

For migration assay, a 200-μl collagen I/Matrigel mix at a concentration of ~4.5 mg/ml collagen I (354249; Corning), and 2 mg/ml Matrigel (354234; Corning) was added to an eight-well glass-bottomed cell culture slides (IBIDI) and let to polymerize at 37°C for 1 h (73, 74). A431 stably expressing Lifeact-GFP were plated on the collagen–Matrigel at a concentration of 10,000 cells/well and cultured for 24 h. Slides were transferred into live-cell imaging mount on an inverted LSM700 confocal microscope (Zeiss) to maintain 5% $CO_2$ and 37°C during movie acquisitions. Time-lapse of 5 min interval were acquired with a 20× Plan Apo, NA 0.8, DICII objective, using Zen software. Clusters with ~6–15 cells were tracked manually by using the ImageJ (75) plugin "Manual Tracking," and tracking was stopped when clusters merge with each other. The recorded x/y position was analyzed using the chemotaxis tool from IBIDI to calculate accumulated distance and velocity (https://ibidi.com/chemotaxis-analysis/171-chemotaxis-and-migration-tool.html). Any tracking with less than 5 h was excluded from accumulated distance calculation.

For periphery displacement and extension/retraction velocities calculation, A431 cells stably expressing Lifeact-GFP were plated on four-well glass-bottomed cell culture slides (IBIDI) and cultured for 24 h. Slides were transferred into live-cell imaging mount on a Leica SP8 confocal fluorescence microscope (Leica Microsystems) to maintain 5% $CO_2$ and 37°C during movie acquisitions. A z-scan of three planes of clusters containing 6–15 cells was performed within a 10 min interval time. Acquisition was made with a 40x/1.3 Plan Apo DIC, using LasX software. Images were processed on the ImageJ software using the "Sum Intensity Z-projection." Cluster periphery detection and extension/retraction velocities were calculated using the ADAPT plugin on ImageJ, created by 76. For periphery displacement calculation, the "velocity visualization" output containing the detected periphery for each timepoint was opened sequentially in ImageJ and merged in a stack. The stacked image was centered using the "Template Matching" plugin created by 77, and projected using the "z-project" tool. By using the "straight line" tool, we measured the thickness of the cluster border in six different positions, which represents the movement of the cluster

periphery over time. The velocity values were directly generated by the ADAPT plugin and plotted in GraphPad Prism.

For eGFP–paxillin or eGFP–MAP4K4 time-lapse acquisition, A431 cells stably expressing eGFP–paxillin or eGFP–MAP4K4 were plated on a four-well glass-bottomed cell culture slides (IBIDI). Cells were transferred into live-cell imaging chamber mount on a Zeiss LSM880 confocal to maintain 5% $CO_2$ and 37°C during movie acquisitions. Time-lapse was acquired with a ×63/1.4 Plan Apochromat oil immersion objective, using Zen software. An interval time of 30 s was used for eGFP–paxillin, whereas an interval time of 15 s was used for eGFP–MAP4K4. For focal adhesion dynamics analysis, we used ImageJ to binarize and threshold the paxillin signal. Increases or decreases in focal adhesion areas was quantified during 30 min, being relativized every three frames. The values presented are the ratio of mean increase, decrease, or stable areas over the sum of the area of these three parameters.

For FRAP experiments, A431 cells stably expressing E-cadherin-mRuby3 were plated on four-well glass-bottomed cell culture slides (IBIDI) and cultured for 24 h. Slides were transferred into live-cell imaging mount on Zeiss LSM880 confocal to maintain 5% $CO_2$ and 37°C during acquisitions. Time-lapse was acquired with a ×63/1.4 Plan Apochromat oil immersion objective, using Zen software. Photobleaching was done with 20 iterations at 100% laser power. Recovery was measured by scanning a region of 40 × 40 pixels (3.92 × 3.92 $\mu m$) at the cell-cell adhesion every 5 s for 10 s pre-bleach and 115 s post-bleach. We performed FRAP experiments on at least five cells per condition. FRAP analysis was performed using the Frapbot software ([78]).

### Protein extraction and immunoblotting analysis

Cells were rinsed in PBS and lysed in lysis buffer (1 M Tris–HCl, pH 7.4, 1% SDS) supplemented with 100 mM PMSF and the protease inhibitor cocktail (11697498001; Sigma-Aldrich) on ice. Cell lysates were centrifuged for 30 min at 4°C and the supernatant was collected. Protein concentrations were calculated using the BCA Protein Assay Kit (Thermo Fisher Scientific). Proteins were separated using SDS–PAGE gels (8%, 10%, or 12%, according to protein size) and transferred into PVDF membranes. The membranes were blocked in skim milk 5% for 1 h and exposed to the primary antibodies diluted in TBS-Tween 0.1%, BSA 2% overnight at 4°C. Secondary antibodies were incubated at RT for 1 h, and membranes were revealed in an X-ray film in a dark room.

### Immunofluorescence

Cells were plated on coverslips to reach cluster confluence as described before. Cells were fixed with 4% PFA for 20 min at RT. Exceptionally, for pERM, staining requires fixation with 10% TCA on ice. Cells were rinsed 3x with PBS and permeabilized with 0.1% Triton X-100 for 2 min. After rinsing 3x with blocking solution (2% BSA in PBS), the coverslips were incubated in blocking solution for 1 h at RT. Primary antibodies were diluted in blocking solution and incubated for 1–3 h at 37°C or overnight at 4°C, both in a humidified chamber. Secondary antibodies were co-incubated with phalloidin and DAPI for 1 h at RT, diluted in blocking solution. The slides were

mounted using Mowiol mounting medium or Vectashield (H-1000; Vector Laboratories).

### Image acquisition and processing

Images were acquired using the confocal microscopes: LSM700 (Carl Zeiss), coupled to a ×63/1.4 Plan Apochromat DIC oil immersion objective, LSM880 (Carl Zeiss), equipped with a ×63/1.4 Plan Apochromat oil immersion objective, or a Leica SP8 (Leica Microsystems), equipped with 63x/1.4 Plan Apo DIC immersion oil objective.

Cluster circularity was calculated using the "shape descriptor" measurement from ImageJ, using the "freehand selection" tool to draw the cluster as the ROI (region of interest). Protrusion areas were calculated also using the "freehand selection" tool from ImageJ. We considered as a protrusion the actin area in front of the nucleus that do not present retraction fibers and were constrained laterally by the actin arches for DMSO, or cell–cell junctions for GNE-495–treated cells. Clusters stained with phalloidin and DAPI were used for those analyses.

For quantifying the F-actin perpendicularly inserted at cell–cell junctions, we recorded a z-scan comprising the entire cell–cell junction, using $\beta$-catenin staining as a reference, and phalloidin for F-actin. Images were recorded every 0.21 $\mu m$ and processed by the AiryScan module on Zen. Analysis was performed using "maximum intensity" projection (ImageJ). Using ImageJ, we plotted the intensity profile of a line scan manually traced along the side of the $\beta$-catenin staining. A minimum F-actin intensity threshold was chosen for each experiment. Each intensity maxima above this threshold was considered as a bundle of F-actin and counted. Intensity maxima were double checked with image to avoid counting non filamentous structures. A total of 15 cell–cell junctions were analyzed from 15 clusters of three independent experiments.

Similar quantification was applied to the F-actin network analysis. For this, we made a z-scan of the entire cluster stained with phalloidin and DAPI. Images were recorded every 0.3 $\mu m$. The first plane was used to measure ventral fibers, whereas the dorsal actin network was determined on "maximum intensity" projections (ImageJ) from the focal plane at the middle of the nucleus to the top of the cluster. Using ImageJ, we plotted the intensity profile of a line scan manually traced from the distal to the proximal part of protruding cells, considering at least three cells per cluster. The same approach described on the previous paragraph was used to quantify the intensity maxima.

Dorsal stress fibers analysis was performed using a z-scan of entire clusters stained for F-actin and $\alpha$-actinin, with a z-interval of 0.3 $\mu m$. The 3D reconstitution of the clusters was performed using Imaris software (Bitplane), and the F-actin structures enriched in $\alpha$-actinin and elongating from the bottom of the cluster to its dorsal part were counted. The estimated length in 3D was also measured manually using Imaris (Bitplane).

Mature focal adhesions number was calculated using the Surface tool on Imaris (Bitplane), from clusters stained for zyxin and F-actin. A minimum threshold of 3 $\mu m^2$ surface was set to also select the mature focal adhesions by size.

Mean intensity of pERM staining was measured using ImageJ by using the "freehand selection" tool to draw the cluster as the ROI.

Number and length of retraction fibers were calculated from clusters stained for pERM. We measured the length manually by using the "segmented line" tool on ImageJ. Background intensity was systematically subtracted.

Intensity of pMLC2 was measured using a z-scan of entire clusters, with a z-interval of 0.3 $\mu$m. "Sum intensity" projection was performed on ImageJ. Cluster and protrusion ROI were selected as mentioned before, using the phalloidin and DAPI channels. The mean intensity and the area of each ROIs were measured and multiplied to calculate the total intensity. Background intensity was systematically subtracted.

Cell–cell junction tortuosity was measured using the "maximum intensity" projection of z-scan of entire clusters stained for p120-catenin. Images were taken every 0.3 $\mu$m. The junction signal at one side of a cell–cell junction was outlined using the "segmented line tool" on ImageJ. Then, we measured the length between the two extremities of this outline using a straight line. The tortuosity was calculated using the ratio of the outline length over the straight length.

For vinculin accumulation at the cell–cell junction, we used the p120 channel to select the ROI, and vinculin colocalization was calculated using the ImageJ "colocalization test" tool, selecting the Pearson's colocalization output.

Line scans for measuring MAP4K4 localization at cell–cell junctions were performed using the "Sum intensity" projection tool from ImageJ, with a z-scan containing the high of the cell–cell junction. Images were taken every 0.7 $\mu$m.

### Cell detachment and scattering assays

For detachment event analysis, A431 cells were infected with pLVpuro–CMV–N-EGFP–MAP4K4 and plated into four-well glass-bottomed cell culture slides (IBIDI) to induce cluster formation, as described before. Cells were transferred into live-cell imaging chamber mounted on a Zeiss LSM880 confocal to maintain 5% $CO_2$ and 37°C during movie acquisitions. Time-lapse was acquired with a 40x/1.4 Plan Apo DIC oil immersion objective, using Zen software. Images were acquired during 2 h, with an interval time of 1 min. A detachment event was considered every time a cell completely detaches from all its neighbors, becoming isolated. Control cells and eGFP–MAP4K4-infected cells were compared. A total of 45 movies for each condition, from three independent experiments were analyzed.

For cell-scattering assay, A431 cells were infected with pLVpuro–CMV–N-EGFP–MAP4K4 (WT, kinase dead, or deleted for the CNH domain) and plated on coverslips to induce cluster formation, as described before. Cells were stained for F-actin and nuclei. Using the LSM880 (Carl Zeiss), coupled to a 20x/0.8 Plan Apo DIC, we acquired tiles of 2 × 2 in eight different regions of the coverslip, randomly selected. The total number of cells per field was determined using the Spot tool on Imaris (Bitplane) to detect the nuclei. The mean eGFP intensity was calculated using the Surface tool on Imaris (Bitplane) on the F-actin channel to detect the total cell area as the ROI. The total eGFP intensity was calculated by multiplying the mean intensity by the cells' area. Background intensity was systematically subtracted.

### Synthesis of polydimethylsiloxane (PDMS) silicone substrates

Compliant PDMS substrates of known stiffness were prepared as described previously (79, 80). To summarize the manufacture, parts A and B of NuSil 8100 (NuSil Silicone Technologies) were mixed at a 1:1 w/w ratio. The stiffness of the silicone substrates was then tuned by adding a certain concentration of Sylgard 184 PDMS cross-linking agent (dimethyl, methyl hydrogen siloxane, containing methylterminated silicon hydride units) to the PDMS. The mechanical properties of these PDMS substrates have previously been extensively characterized (79, 80). For our experiments, we selected a concentration of 0.36% w/w Sylgard 184 crosslinker, resulting in a 12 kPa stiffness as this is within the range of in vivo stiffness epidermoid carcinoma and resembles a stiffer tumor microenvironment. 50 $\mu$l of uncured PDMS, spread onto square ~22 mm (no. 1) glass coverslips was cured for 1 h at 100°C to yield silicone substrates with a 100-$\mu$m thickness. DID-conjugated (far-red) fluorescent fiduciary beads were synthesized as described previously (81), mixed into uncured PDMS and crosslinker, then spin-coated onto the PDMS substrates at 3,000 rpm for 1 min to yield a bead-embedded PDMS layer ~1 $\mu$m thick (WS-650 Spin Processor; Laurell Technologies). The substrates were then incubated at 100°C for 1 h. They were then fastened to the bottom of six-well plates, functionalized using sulfo-SANPAH, and then protein coated with collagen for cell adhesion.

### TFM and BISM

Cell-generated surface displacements, traction stress, and strain energy were quantified using TFM as described in the literature (79, 82) using an open-source Python TFM package modified to the case of cell clusters and force imbalances within the field of view (35, 83, 84). Intercellular stresses, shear, and normal stresses were then quantified from the traction forces using BISM using a MATLAB package (85). Cells were plated, allowed to settle and form clusters 48 h before imaging. Before imaging, the cells were treated with Hoechst 33342 dye (Thermo Fisher Scientific, Inc.) to stain the nuclei. During imaging, A431 cells expressing mCitrine-tagged P-cadherins and A431 cells expressing mRuby-tagged E-cadherins were used. We focused on imaging isolated cell clusters in a clear field of view consisting of 5–13 cells. For imaging, the six-well plates containing the cells were mounted onto a confocal microscope (Leica TCS SP8 with a 10 Å~0.4 NA objective). They were then maintained at 37°C (stage heater; Cell MicroControls) and 5% $CO_2$ (perfusing 100% humidity pre-bottled 5% $CO_2$ in synthetic air). The cells, nuclei, and fiduciary TFM beads were simultaneously imaged using fluorescent and transmission microscopy over several hour time courses at time intervals of 15–40 min. The resulting images were then corrected for lateral drift using an ImageJ pipeline, the values outside the cell cluster were masked to remove the background noise, then analyzed using the Python TFM and BISM workflows to obtain the displacements, tractions, strain, and intercellular stresses.

## Statistical analysis

All graphs and statistical tests were performed using GraphPad Prism (GraphPad Software). We performed at least three independent experiments (N) for each analysis, and the minimum number of data points (n) is specified at figure legends. Because normal distribution of the data was not formally tested, we used the non-parametric Mann–Whitney test or Kruskal–Wallis test with Dunn's correction for multiple comparisons of unpaired data. Paired data were analyzed using Wilcoxon test. Values are expressed as mean ± s.d., unless otherwise indicated at the figure legend, and all individual values are represented at the graphics. $P$-values are noted on figures as following: $*P < 0.05$, $**P < 0.01$, $***P < 0.001$, $****P < 0.0001$.

## Supplementary Information

## Acknowledgements

We thank Arnold Hayer (McGill U), Sebastien Carréno (U Montreal), and Weilan Ye (Genentech) for their generosity in sharing reagents. We thank C Charbonneau for technical assistance and the entire Emery laboratory and Arnold Hayer for helpful discussions and critical reading of the manuscript. This work was supported by grants from the Canadian Institute for Health Research (CIHR; PJT—175093 to G Emery and AJ Ehrlicher, and PJT-143327 to AJ Ehrlicher), from the Natural Sciences and Engineering Research Council of Canada (NSERC, RGPIN-2020-07169 to AJ Ehrlicher) and from Canadian Foundation for Innovation (Project #32749 to AJ Ehrlicher). LE Alberici Delsin held a doctoral scholarship and C Plutoni a postdoctoral fellowship from Fonds de Recherche du Québec—Santé (FRQS).

### Author Contributions

LE Alberici Delsin: conceptualization, data curation, formal analysis, validation, investigation, visualization, methodology, project administration, and writing—original draft, review, and editing.
C Plutoni: conceptualization, formal analysis, validation, investigation, and methodology.
A Clouvel and S Keil: formal analysis, validation, investigation, and methodology.
L Marpeaux and L Elouassouli: validation and investigation.
A Khavari: validation, investigation, and methodology.
AJ Ehrlicher: data curation, supervision, funding acquisition, and methodology.
G Emery: conceptualization, resources, data curation, formal analysis, supervision, funding acquisition, methodology, project administration, and writing—review and editing.

### Conflict of Interest Statement

The authors declare that they have no conflict of interest.

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
