## [Reviewer comments · Life Science Alliance]

MAP4K4 regulates forces at cell-cell and cell-matrix adhesions to promote collective cell migration

Lara Alberici Delsin, Cedric PLUTONI, Anna Clouvel, Sarah Keil, Léa Marpeaux, Lina Elouassouli, Adele Khavari, Allen Ehrlicher, and Gregory Emery

DOI: <https://doi.org/10.26508/lsa.202302196>

Corresponding author(s): Gregory Emery, Institute for Research in Immunology and Cancer

Review Timeline:	Submission Date:	2023-06-02
	Editorial Decision:	2023-06-06
	Revision Received:	2023-06-12
	Accepted:	2023-06-16

Transaction Report:

Please note that the manuscript was previously reviewed at another journal and the reports were taken into account in the decision-making process at Life Science Alliance.

Reviewer #1 Review

Comments to the Authors (Required):

This manuscript assesses the role of MAP4K4 in collective cancer cell migration in A431 cells. They report, in line with previous studies, that CRISPR knockout of MAP4K4 induces an increase in cell spreading, a decrease in focal adhesion turnover and increased actin stress fibers and traction force, and higher tensile stress at junctions. MAP4K4 also regulates phosphorylation of moesin, knockdown of which gives similar effects, from which they conclude that moesin phosphorylation mediates. They also see localization to cell-cell junctions and conversion of junctions from linear to "tortuous", indicative of lower stability. They report that deletion of the the C-terminal domain ablates localization to junctions and partially weakens the effect on junctional morphology. Based on these results, they conclude that MAP4K4 regulates focal adhesions through moesin phosphorylation and cell-cell junction stability through an unknown mechanism.

There are a number of points of interest here but overall the main conclusions are not convincing, due to a combination of specific experimental weakness and failure to consider alternative hypotheses. As noted in the manuscript, previous work has shown that MAP4k4 promotes focal adhesion turnover. Extensive studies have also shown crosstalk between focal adhesions, tension and cell-cell adhesions such that strong focal adhesions and high cytoskeletal tension destabilizes junctions (for example, PMID 16216928, or see PMID 29454273 for a review). At present, a mechanism in which increased cell spreading increases cytoskeletal tension, which destabilizes cell-cell junctions offers the simplest explanation of their results. Another general weakness is that nearly all of the work is done in a single cell line studying migration in 2D. There is no attempt to establish relevance to collective cell migration in vivo or even in 3D.

Specific Criticisms:

1. Figs 1 and 2. The experiments that use test the inhibitor in MAP4K4 KO cells. If it is specific, there should be no further effect in these cells.
2. For Fig 1h, I, it is not explained what pre-treatment means. Washout?
3. Multiple figures. Traction force and stress fibers increase as a function of spread cell area (PMID 21163521, 12112152 among others). A central question for this study is whether the increase in stress fibers and traction force is simply due to greater cell spreading in MAP4K4 KO or inhibited cells. The most rigorous way to do this is to set cell area using engineered substrates.
4. Fig 3a. focal adhesion lifetime/turnover needs to be quantified.
5. Fig 3d,e. Phospho-ERM needs to be assayed by Western blotting as well as staining to assess whether the observed

difference is due to decreased phosphorylation, decreased localization at the basal plane, or decreased accessibility to the antibody.

6. Fig 3f. Why are retraction fibers seen in this panel but not any of the other panels in the paper?

7. Fig 3g-i. The ERM knockdowns need rescue controls including rescue with WT vs phospho-mutant moesin.

8. Fig 6a,b, video. The authors conclude that recruitment of MAP4K4 to cell-cell junctions precedes junction disassembly based on anecdotal evidence without any attempt at more rigorous analysis.

9. Fig 6g,h. The deltaCNH mutant that does not localize to cell-cell junctions has a weaker effect on junction morphology than WT, which is interpreted to indicate that localization to junctions is required for this effect. But the ability of this mutant to affect stress fiber organization and contractility is not assessed. This mutation may equally affect focal adhesions and contractility, which would largely undermine this conclusion.

10. The authors conclude: "At focal adhesions, MAP4K4 seems to act predominantly through Moesin phosphorylation. However, our data suggest that this is not the case at adherens junctions." No evidence is presented that moesin does not also mediate the effects on cell-cell junctions. But if it is correct, that analysis would significantly strengthen their case.

Reviewer #2 Review

Comments to the Authors (Required):

In this work, the role of MAP4K4 in the collective migration of A431 clusters is unveiled. At the cellular level, MAP4K4 impinges on the dynamics of cell protrusions, presumably by regulating the remodeling of the actin cytoskeleton. It also influences the stability of both cell-cell and cell-substrate adhesion. Indeed, MAP4K4 loss of functions stabilizes focal adhesion through Moesin phosphorylation, but also adherens junctions independently on Moesin activity. Using traction forces microscopy, it is shown that the loss of MAP4K4 increases the traction forces on the substrates and the tensile stress on junctions. MAP4K4 is also shown to be recruited at the cell-cell junction and to modulate junctional dynamics through, however, ill-defined mechanisms. Thus, MAP4K4 appears central in regulating cellular and supracellular force balance during collective motility.

The work is generally carefully executed with a set of straightforward experiments that support most of the conclusions of the paper. The first part of the manuscript (From Figure 1 -to 4) is largely confirmatory of previous work in different systems some of which were performed by the same authors. The most novel aspect is in the characterization of the potential role that MAP4K4 might exert in cell-cell junction morphology, dynamics, and stability. In this respect, however, the data presented are interesting but underdeveloped and do not provide a clear picture of the mechanistic or functional role of MAP4K4 at cell-cell junctions. For example. It is unclear how junctional tortuosity is affected by the LOF of MAP4K4, or how MAP4K4 destabilizes the junction (or increases junctional dynamics). Some evidence is provided that MAP4K4 does so independently of its function on FA, but the key experiment supporting this is the lack of effect on the junction of the LOF of Moesin. In summary, more work seems to be needed to clarify the novel role of MAP4K4 in the control of junctional biology.

Several specific points would also need to be clarified.

1. It is shown that pharmacological or genetic interference with MAP4K4 impairs the motility of A431 clusters. Analysis of protrusion dynamics within cells in the clusters further shows impairment in protrusion extension and retraction. (Figure 1). It is unclear whether the effect of protrusion dynamics is an emergent property of cell collectives or, alternatively, MAP4K4 also impacts the cell motility and protrusion dynamics in isolated, individual cells or cells, plated individually, along microprinted tracks.

2. Minor: In figure 1, CRISPR technology is used to generate clones of the A431 Devoid of MAP4K4. Can the migratory phenotype be restored by re-expression of MAP4K4? It must be pointed, out that the authors have used also different inhibitors(Extended Data Fig. 2a) of MAP4K4 that all seem to give rise to the same phenotype as the genetic LOF. Yet the availability of CRISPR clones could have been exploited in structure function reconstitution studies that monitored not only junctional morphology (as it is shown in Figure 6) but migratory parameters.

3. In Figures 2 and 3, it is shown that LOF of MAP4K4 results in increased cell spreading presumably mediated by the stabilization of focal adhesion dynamics, which, in turn, influence transverse actin arc formation, cell spreading areas, and protrusion dynamics. All these effects appear to be mediated by Moesin, the phosphorylation of which is impaired as previously shown in other systems.

Fig. 4 documents an increase in traction forces on the substrate in LOF of MAPK4, again presumably as a consequence of the altered FA dynamics.

Comments: Fig. 2 -4 are technically well executed, but by and large, reproduce findings obtained previously. It remains unclear what the novelty of the findings reported is. Are all the phenotypes described recapitulated by the effect of MAP4K4 on FA dynamics? Stated differently why the work is well executed and carefully conducted. It appears a confirmatory study.

4. The effect of LOF of MAP4K4 on junctional morphology and tensile stress is interesting and certainly more novel, particularly since the LOF of Moesin does not affect this phenotype. There are a few things to clarify. The increased tortuosity of the cell-cell junction might indicate less longitudinal tension (which could and probably should be measured directly in various ways) or, as suggested, by an increase in perpendicular contractility, mediated presumably by elevated actomyosin contractility. However, staining with pMLC indicated that the elevation of its signal is mainly confined to cell protrusions. Hence it remains unclear how the junction tensile state is brought about by the LOF of MAP4K4

5. Some answers to the last point are tentatively offered in the last set of experiments in figure 6. Here, MAP4K4 is shown to be transiently and dynamically recruited to the cell-cell junction. It is not clear, however, what MAP4K4 does there. The authors

6. Of note, the author, in Figure 6e, used a structure-function reconstitution approach to investigate the impact of critical domains of MP4K4 on junctional morphology. It would seem necessary to complement this experiment with an analysis of cluster dynamics and migration.

Reviewer #3 Review

propose that MAP4K4 promotes junctional turnover and, consistent with this idea, they show that the ectopic expression of the proteins enhances cell scattering. However, it would seem necessary to directly study the impact of MAP4K on junctional stability by directly measuring, for example, the turnover of junctional proteins (such as E-cadherin) through, FRAP experiments. Some clues as to the molecular mechanisms through which MAP4K4 operates at the cell-cell junction would also seem required (see general comments).

Here is the revised version of our manuscript. As per our previous exchanges, we have focused our revisions on specific points raised by the reviewer, and in addition, we have tried to answer to the vast majority of their comments.

The principal changes are:

- 1) We now describe the localization of MAP4K4 with high spatio-temporal resolution.
- 2) We also improved the characterization of actomyosin at cell-cell junctions.
- 3) We provide a detailed quantification of the phenotype induced by overexpressing MAP4K4.
- 4) We have reorganized the results to improve clarity and we have rewritten large parts of the discussion and of the result section to better align our results and their interpretation, and to better highlight the novelty of our work.

As per request, we also determined if the mobility of E-cadherin is changed in gain and loss of function of MAP4K4. We did not observe a change, but, as described below, the analysis by FRAP experiment cannot be performed on dynamic junctions, hence the results might not be representative of the impact of MAP4K4 overexpression on E-cadherin dynamics.

We think that overall, we have reinforced the message of our manuscript, and that the novelty of our study is better outlined in the revised version. Below is point-by-point answer to the reviewers' comment.

We hope that you will agree that the manuscript is strikingly improved and that our conclusions are important, and worth being published in a high-quality journal.

Reviewer #1

General comments:

This manuscript assesses the role of MAP4K4 in collective cancer cell migration in A431 cells. They report, in line with previous studies, that CRISPR knockout of MAP4K4 induces an increase in cell spreading, a decrease in focal adhesion turnover and increased actin stress fibers and traction force, and higher tensile stress at junctions. MAP4K4 also regulates phosphorylation of moesin, knockdown of which gives similar effects, from which they conclude that moesin phosphorylation mediates. They also see localization to cell-cell junctions and conversion of junctions from linear to "tortuous", indicative of lower stability. They report that deletion of the the C-terminal domain ablates localization to junctions and partially weakens the effect on junctional morphology. Based on these results, they conclude that MAP4K4 regulates focal adhesions through moesin phosphorylation and cell-cell junction stability through an unknown mechanism.

There are a number of points of interest here but overall, the main conclusions are not convincing, due to a combination of specific experimental weakness and failure to consider alternative hypotheses. As noted in the manuscript, previous work has shown that MAP4k4 promotes focal adhesion turnover. Extensive studies have also shown crosstalk between focal adhesions, tension and cell-cell adhesions such that strong focal adhesions and high cytoskeletal tension destabilizes junctions (for example, PMID 16216928, or see PMID 29454273 for a review). At present, a mechanism in which increased

cell spreading increases cytoskeletal tension, which destabilizes cell-cell junctions offers the simplest explanation of their results. Another general weakness is that nearly all of the work is done in a single cell line studying migration in 2D. There is no attempt to establish relevance to collective cell migration in vivo or even in 3D.

We thank the reviewer for their general assessment of our manuscript, although we respectfully disagree with some of their conclusions. In particular, the reviewer indicates that we concluded that “the conversion of junctions from linear to tortuous” is “indicative of lower stability”. We assume the reviewer made this statement based on the literature, specially characterized on endothelial cells, where similar morphological changes lead to junction disassembly [1-3]. However, our data indicate that this interpretation is not adequate in our case. Indeed, we found that cell scattering (a consequence of junction disassembly) is promoted by MAP4K4 overexpression, while cell tortuosity was due to MAP4K4 inhibition. Our conclusion is thus opposite to what was previously described. We think that this divergence has led to several misinterpretations of our data and conclusions (see thereafter).

Furthermore, this summary, in our opinion, minimize the novelty of our work and overlook our data regarding the overexpression of MAP4K4, that we regard as important in the context of the role of MAP4K4 in cancer progression. This may be due to the organization of our previous version of our manuscript, and we hope that the reviewer will be more convinced with the revised version. We have reorganized large parts of the manuscript so that the relevance of our findings is better outlined.

We also disagree that “increased cell spreading increases cytoskeletal tension, which destabilizes cell-cell junctions” would explain our results. Again, this comment seems to be based on the literature, with experiments performed on other cell types rather than on the data presented in the manuscript. Indeed, we show that inhibition of MAP4K4 leads to spreading and stable junctions, while overexpression of MAP4K4 leads to more dynamic junctions (without spreading). Hence, according to the reviewers’ logic, our data precisely indicate that the roles of MAP4K4 on focal adhesion and cell-cell junction are independent. Accordingly, the KO of moesin, that phenocopies MAP4K4 loss-of-function, induces the same spreading phenotype but does not induce tortuous junctions, showing that the impact on focal adhesion can be dissociated from the phenotype at cell-cell junctions. Hence, we stand with our conclusion that MAP4K4 acts through a different mechanism at focal adhesions and at cell-cell junctions.

To outline this finding and to take in account several papers that explore the crosstalk between focal adhesions and adherens junctions, and their positive or negative correlation upon increase of tension depends on cell context, we wrote the following in the discussion section:

“Focal adhesions and adherens junctions are indirectly connected through the cell cytoskeleton and several proteins are shared between the two structures. The mechanical crosstalk between them has been explored and they can present cooperating or antagonistic responses. Here, we report a functional mechanism where MAP4K4 activity is central for balancing traction force generation through disassembly of focal adhesions, while it is also required to decrease the intercellular stresses and tension loading at the adherens junctions.”

We think that studying collective cell migration in 2D is very relevant and such studies are widely performed by numerous laboratories. In particular, 2D cell migration gives us the opportunity to measure biomechanical properties that are difficult or even impossible to assess in 3D or in vivo. Finally, we have previously characterized the role of the MAP4K4 ortholog Misshapen in the collective migration of border cell, a powerful in vivo model to study collective cell migration [4]. Also, note that Misshapen has been involved by others in various collective migration in Drosophila [5]. For our study, we have precisely selected a model that shares some features of the migration of border cells to determine the similarities and the differences of the role of an orthologue of Misshapen in collective cell migration of small cell clusters.

Specific Criticisms:

1. Figs 1 and 2. The experiments that use test the inhibitor in MAP4K4 KO cells. If it is specific, there should be no further effect in these cells.

This is not obligatory the case as in absence of its main target, the drug may affect other kinases. Nevertheless, we are very confident about our findings. The drug is quite specific as was described in several publications [6, 7]. We used other drugs, equally specific [8-10] to validate our findings and the KO of MAP4K4. This exceed what is performed in most studies. In addition, we tested siRNA for all three Misshapen orthologs (MAP4K4, MINK and TNIK) and we found that the siRNA for MAP4K4 is the only one reproducing the phenotype induced by the KO and the inhibitors. Note that the siRNA experiments were not used extensively as siRNA treatments stress A431 cells. For this reason, we decided not to include these results in the revised manuscript.

2. For Fig 1h, I, it is not explained what pre-treatment means. Washout?

We agree with the reviewer that “pre-treatment” is not intuitive. In pre-treatment condition, the cells are in regular medium, and we add the GNE compound (or DMSO) when indicated. We changed the wording to “before treatment (1h)”.

3. Multiple figures. Traction force and stress fibers increase as a function of spread cell area (PMID 21163521, 12112152 among others). A central question for this study is whether the increase in stress fibers and traction force is simply due to greater cell spreading in MAP4K4 KO or inhibited cells. The most rigorous way to do this is to set cell area using engineered substrates.

The reviewer raises an excellent point that cell spread area and traction force are generally thought to be correlated. They suggest that us establishing cause and effect between the two is a central question of the manuscript, specifically that we must investigate if increased contractility is a product of cell spreading. We suggest that this is not a question for the manuscript for the following reasons: 1) Our results focus on the wholistic changes of contractility in response to MAP4K4 changes, regardless of underlying changes in aspects such as focal adhesion density or cell shape. 2) These parameters are indeed linked, and diverse numerous studies have examined this question in detail coming to a consensus that it is in fact a cyclic feedback loop that starts with nascent focal adhesion formation, followed by actomyosin tension, followed by cell spreading and adhesion maturation. 3) Confining cells to an artificial geometry would serve to artificially constrain the max contractile stress and work exerted by the cell [11] 4) it is a cumbersome experiment to confine cells which does not reveal more about the nature of MAP4K4. 5) we report the strain energy as well, demonstrating that MAP4K4 cells not only exert larger stresses but also higher contractile work which is not dependent on cell area.

4. Fig 3a. focal adhesion lifetime/turnover needs to be quantified.

The data are now provided in Extended Data Fig. 3b.

5. Fig 3d,e. Phospho-ERM needs to be assayed by Western blotting as well as staining to assess whether the observed difference is due to decreased phosphorylation, decreased localization at the basal plane, or decreased accessibility to the antibody.

We have performed a WB for total pERMs and found no significantly changes. We did not pursue this observation for several reasons: 1) The different pERM cannot be separated in WB., 2) ERMs are present at the cortex and apical sides of the cells and these pools are not regulated by MAP4K4, 3) As a consequence, we reasoned that it would be more significant to monitor ERMs at focal adhesions and at adherens junctions. These data are provided in the manuscript (Extended Data Fig.5a-f). 4) We performed the KO of ezrin, radixin and moesin and found that the moesin-KO reproduced the MAP4K4 phenotype at focal adhesions. 5) These data are consistent with what was shown by others [7].

Results regarding ERM phosphorylation and Moesin KO were reorganized into a single figure (Extended Data Fig. 5) to improve clarity, and text was modified accordingly:

“In endothelial cells, MAP4K4 disassembles focal adhesions through the local phosphorylation of Moesin. Moesin is a member of the ERM (Ezrin, Radixin, Moesin) family of proteins that link cortical actin to the plasma membrane. Phosphorylation of Moesin by MAP4K4 ultimately induces integrin inactivation and focal adhesion disassembly. To test if a similar mechanism is at play in A431 cells, we first monitored phosphorylated ERM (pERM) using a phospho-specific antibody that recognizes a conserved phosphosite in all ERM proteins. We observed that MAP4K4 inhibition decreases pERM intensity at the focal plane containing the focal adhesions (Extended Data Fig. 5a, b). Concomitantly, we observed a reduction in the number and length of retraction fibers, which are ERM enriched structures that are formed when cells retract (Extended Data Fig. 5c, d). Surprisingly, pERM intensity measured at the adherens junction focal plane did not significantly after MAP4K4 inhibition (Extended Data Fig. 5e, f).

To understand if these effects are due to loss of Moesin phosphorylation, we generated Moesin KO cells by CRISPR-Cas9, using two independent guide sequences (sgRNA) (Extended Data Fig. 5g). The Moesin KO cells presented an increase in zyxin-enriched mature focal adhesions (Extended Data Fig. 5h, i), suggesting that MAP4K4 phosphorylates Moesin to regulate the dynamics of focal adhesion also in A431 epidermoid cells. However, Moesin KO clusters do not present tortuous junctions (Extended Data Fig. 5j, k), showing that the loss of Moesin is not sufficient to phenocopy the MAP4K4 LOF effect at adherens junction.

Altogether our data suggest that MAP4K4 acts on a different substrate at adherens junction. Moreover, because Moesin KO increases the number of mature focal adhesions, but does not make cell-cell junctions more tortuous, we can hypothesize that the effect of MAP4K4 LOF at cell-cell junctions is not an indirect effect of focal adhesion stabilization. Hence, MAP4K4 might directly regulate forces at adherens junctions.”

6. Fig 3f. Why are retraction fibers seen in this panel but not any of the other panels in the paper?

In addition to Fig. 3f, retraction fibers can be seen in 1d (indicated by arrowheads), Extended Data Fig. 4e, Extended Data Fig. 6a and 6b.

7. Fig 3g-i. The ERM knockdowns need rescue controls including rescue with WT vs phospho-mutant moesin.

These experiments are not easy to perform as expression levels are hard to control and A431 cells are finicky to transfect. Also, we respectfully disagree with the reviewer that the rescue is necessary: we show clearly that Moesin is gone in the KO cells. Furthermore, as described above and in the manuscript, the phenotype is consistent with what was previously published. Finally, for Moesin to function properly at focal adhesion, it needs to cycle between an inactive and an active state, hence we do not expect that using phospho-mutants will be conclusive.

8. Fig 6a,b, video. The authors conclude that recruitment of MAP4K4 to cell-cell junctions precedes junction disassembly based on anecdotal evidence without any attempt at more rigorous analysis.

We agree with the reviewer that our analysis of the time-lapse video was not detailed. This observation was the entry point for the analysis of the cell-cell junction disassembly, and although, yes anecdotal, it led to the finding that overexpression of MAP4K4 disassembles the junctions. We now performed higher spatiotemporal resolution imaging of MAP4K4 at cell-cell adhesions, combining E-cadherin-mRuby expression, that was used as marker of adherens junction to increase confidence on MAP4K4 localization. Results were added as time-lapse images (Fig. 6a) and supplementary Videos (2 and 3). Due to the dynamic of the junctions, we could not precisely quantify MAP4K4 recruitment to the junctions, but we quantified the number of detachment events on cells overexpressing MAP4K4. Therefore, we modified the text accordingly:

“To further characterize the role of MAP4K4 at cell-cell adhesion, we performed confocal time-lapse imaging of cells expressing both eGFP-MAP4K4 and E-cadherin-mRuby. Interestingly, we observed that MAP4K4 localizes at the cell-cell interface of detaching cells (Fig. 6a, Video 2, 3). To better understand this cell behavior, we imaged A431 cluster control or expressing eGFP-MAP4K4 during 2h with a 1min time resolution. Cells expressing eGFP-MAP4K4 were frequently found as single cells, and the percentage of cells detaching from clusters was significantly higher in this condition. Therefore, MAP4K4 overexpression induces cell-cell adhesion disassembly and cell detachment.”

9. Fig 6g,h. The deltaCNH mutant that does not localize to cell-cell junctions has a weaker effect on junction morphology than WT, which is interpreted to indicate that localization to junctions is required for this effect. But the ability of this mutant to affect stress fiber organization and contractility is not assessed. This mutation may equally affect focal adhesions and contractility, which would largely undermine this conclusion.

To better understand the relevance of the mutants for MAP4K4 KO phenotype and address the reviewer’s point, we analyzed the number of mature focal adhesion on the rescue experiments. Our results indicate that both kinase dead and deltaCNH constructs were unable to rescue MAP4K4 LOF phenotype (Fig.5e). In our understanding, those results do not undermine our conclusion. Instead, they indicate that, on top of regulating MAP4K4 localization, the CNH domain is necessary for the overall function of MAP4K4.

10. The authors conclude: "At focal adhesions, MAP4K4 seems to act predominantly through Moesin phosphorylation. However, our data suggest that this is not the case at adherens junctions." No evidence is presented that moesin does not also mediate the effects on cell-cell junctions. But if it is correct, that analysis would significantly strengthen their case.

We respectfully disagree with the reviewer. We show that the KO of Moesin does not affect adherens junctions. If Moesin was the target of MAP4K4 at junctions, and since it is well established that Moesin is activated by MAP4K4 [5], Moesin KO cells would display a AJs phenotype similar to the inhibition of MAP4K4. The absence of phenotype is strong evidence that Moesin does not mediate the effect on cell-cell junctions. We reorganized the results regarding ERM phosphorylation and Moesin KO into a single figure (Extended Data Fig. 5) to increase clarity.

Reviewer #2

General comments:

In this work, the role of MAP4K4 in the collective migration of A431 clusters is unveiled. At the cellular level, MAP4K4 impinges on the dynamics of cell protrusions, presumably by regulating the remodeling of the actin cytoskeleton. It also influences the stability of both cell-cell and cell-substrate adhesion. Indeed, MAP4K4 loss of functions stabilizes focal adhesion through Moesin phosphorylation, but also adherens junctions independently on Moesin activity. Using traction forces microscopy, it is shown that the loss of MAP4K4 increases the traction forces on the substrates and the tensile stress on junctions. MAP4K4 is also shown to be recruited at the cell-cell junction and to modulate junctional dynamics through, however, ill-defined mechanisms. Thus, MAP4K4 appears central in regulating cellular and supracellular force balance during collective motility.

The work is generally carefully executed with a set of straightforward experiments that support most of the conclusions of the paper. The first part of the manuscript (From Figure 1 -to 4) is largely confirmatory of previous work in different systems some of which were performed by the same authors. The most novel aspect is in the characterization of the potential role that MAP4K4 might exert in cell-cell junction morphology, dynamics, and stability. In this respect, however, the data presented are interesting but underdeveloped and do not provide a clear picture of the mechanistic or functional

role of MAP4K4 at cell-cell junctions. For example. It is unclear how junctional tortuosity is affected by the LOF of MAP4K4, or how MAP4K4 destabilizes the junction (or increases junctional dynamics). Some evidence is provided that MAP4K4 does so independently of its function on FA, but the key experiment supporting this is the lack of effect on the junction of the LOF of Moesin. In summary, more work seems to be needed to clarify the novel role of MAP4K4 in the control of junctional biology.

We thank the reviewer for their positive assessment of our manuscript.

About the lack of novelty: the reviewer pointed out that increase in traction forces and stress fibers are a consequence of focal adhesion stabilization, which has been reported previously. We agree, but we are the first to provide a detailed characterization of all those phenotypes in MAP4K4 LOF. We believe that it is important to validate experimentally the link between focal adhesions stabilization and traction forces as the relation between both is not always linear (e.g. [11]). The formation of stress fibers could also be defective in the LOF, we showed that they are active and able to generate force. Also, to the extent of our knowledge, we are the first to document precisely that MAP4K4 induces the accumulation of mature focal adhesions, using zyxin marker, further characterizing previously described phenotypes.

We have restructured our manuscript to better highlight the novelty of our work. Figures 1 to 4 (and Extended Data Fig. 1 and 2) from the first submission were reorganized into Figures 1 and 2 (and Extended Data Fig. 1-4). With those modifications, we aimed to emphasize the effects of MAP4K4 LOF on mature focal adhesions and on the cell cytoskeleton, highlighting the comprehensive characterization never performed previously. We also regrouped our traction forces and intercellular stresses data into a single figure (Figure 3), bringing those novel measurements earlier in the story.

“The most novel aspect is in the characterization of the potential role that MAP4K4 might exert in cell-cell junction morphology, dynamics, and stability. In this respect, however, the data presented are interesting but underdeveloped and do not provide a clear picture of the mechanistic or functional role of MAP4K4 at cell-cell junctions.”

We have extended our characterization of the impact of MAP4K4 on actin structures at the level of cell-cell junctions and believe that we provide now sufficient mechanistic insights. Indeed, we describe the functional role of both (1) MAP4K4 inhibition (increase intercellular tensile stress, affecting cell-cell adhesion morphology and molecular composition, in a contractility dependent way) and (2) MAP4K4 overexpression (cell-cell junction disassembly, culminating in cell detachment and loss of collectiveness).

However, at that stage, we were not able to identify the substrate of MAP4K4 at cell-cell junction. Identifying the substrate would require a screen-based approach and targets validation, which is in our opinion beyond the scope of this manuscript. Moreover, we listed the known MAP4K4 targets and discussed them in the cell-cell junction context at our discussion session.

Specific Criticisms:

1. It is shown that pharmacological or genetic interference with MAP4K4 impairs the motility of A431 clusters. Analysis of protrusion dynamics within cells in the clusters further shows impairment in protrusion extension and retraction. (Figure 1). It is unclear whether the effect of protrusion dynamics is an emergent property of cell collectives or, alternatively, MAP4K4 also impacts the cell motility and protrusion dynamics in isolated, individual cells or cells, plated individually, along microprinted tracks.

Despite we haven't evaluated the impact of MAP4K4 loss-of-function on individual A431 cells, several previous works described this phenotype in different cell lines. MAP4K4 inhibition led to a decrease in motility of medulloblastoma individual cells, endothelial cells, and keratinocytes [7, 12, 13], by decreasing integrin endocytosis and focal adhesion turnover, accompanied by increase in cell spreading. In endothelial cells, it was also reported a decrease in retraction fibers on individual cells [7]. Therefore, our goal in this work was to understand the role of MAP4K4 on the collective

context, which included some common features with the phenotype in individual cells, but also emergent roles from the collective context.

Aiming at highlighting the MAP4K4 roles emergent from the collective properties, in addition we modified the following parts of the text:

(Results session, line 160): *“Interestingly, due to the collective properties of our chosen model of study, we were able to observe a reorganization of the cell cytoskeleton near the cell-cell junctions. Specifically, actomyosin fibers from the transversal arches are bound perpendicularly to the cell-cell junction, presenting a continuous organization between cells (Fig. 2 j).”*

(Discussion session, paragraph 1): *“In this work we used the epidermoid carcinoma cell line A431 as a model to study the role of the kinase MAP4K4 in regulating the collective migration of cancer cell clusters in vitro. We show that MAP4K4 is a central regulator of force generation and transmission during collective migration, acting specifically on the disassembly of cell-substrate and cell-cell adhesions. Previous work showed that MAP4K4 regulates the disassembly and recycling of focal adhesions in individual cells, through different molecular mechanisms [7, 12, 13]. Here, we found that MAP4K4, in the collective context, also induces focal adhesion turnover predominantly through Moesin phosphorylation in A431 carcinoma cells. Moreover, we extend our analysis and show that when MAP4K4 is impaired there are more mature focal adhesions, in parallel to an increase in the number of stress fibers at cell protrusions, and a decrease in protrusion dynamics. Therefore, MAP4K4 inhibition not only prevents cells to retract, but reorganizes the cytoskeleton network inside protrusions. Furthermore, we reveal that this process relocates active myosin to those protrusions, locally increasing contractility and inducing cells to exert higher traction forces on the substrate (Fig. 7a). Moreover, we observed emergent properties of MAP4K4 due to the collective context. The presence of protrusions around the entire cluster, as observed in MAP4K4 LOF, impairs the cell-cell coordination mechanism and blocks migration [14]. Furthermore, MAP4K4 LOF induced a continuous F-actin organization through cell-cell junctions, favoring the formation of a dorsal F-actin network, that may influence cell-cell communication. Therefore, in addition to stabilizing focal adhesions, MAP4K4 LOF promotes a cascade of cellular processes that culminates in the impairment of collective cell movement (Fig. 7b, c).”*

2. Minor: In figure 1, CRISPR technology is used to generate clones of the A431 Devoid of MAP4K4. Can the migratory phenotype be restored by re-expression of MAP4K4? It must be pointed, out that the authors have used also different inhibitors(Extended Data Fig. 2a) of MAP4K4 that all seem to give rise to the same phenotype as the genetic LOF. Yet the availability of CRISPR clones could have been exploited in structure function reconstitution studies that monitored not only junctional morphology (as it is shown in Figure 6) but migratory parameters.

Transfection of A431 cells is very difficult to perform and expression through lentiviruses, leads to heterogenous levels of expression. As such, measuring migration might be misleading and we decided to focus on parameters that we can observe in fixed samples, where we can select cluster with high, homogenous GFP expression levels. We added mature focal adhesion quantifications to reinforce our observations.

Comments: Fig. 2 -4 are technically well executed, but by and large, reproduce findings obtained previously. It remains unclear what the novelty of the findings reported is. Are all the phenotypes described recapitulated by the effect of MAP4K4 on FA dynamics? Stated differently why the work is well executed and carefully conducted. It appears a confirmatory study.

Please see our answer to the general statement of the reviewer.

4. The effect of LOF of MAP4K4 on junctional morphology and tensile stress is interesting and certainly more novel, particularly since the LOF of Moesin does not affect this phenotype. There are a few things to clarify. The increased tortuosity of the cell-cell junction might indicate less longitudinal tension (which could and probably should be measured

directly in various ways) or, as suggested, by an increase in perpendicular contractility, mediated presumably by elevated actomyosin contractility. However, staining with pMLC indicated that the elevation of its signal is mainly confined to cell protrusions. Hence it remains unclear how the junction tensile state is brought about by the LOF of MAP4K4

We thank the reviewer for this interesting suggestion. To extend our analysis on perpendicular contractility at cell-cell adhesions we performed higher resolution imaging with enhanced confocal resolution (Airyscan detector, Zeiss). β -catenin, F-actin and pMLC2 were monitored at cell-cell adhesions of control or MAP4K4 inhibited cells. We found that (1) there are more perpendicular F-actin inserted at cell-cell adhesion in MAP4K4 LOF than control cells (Fig. 2i) (2) those perpendicular F-actin accumulates all along the junction, as shown in the z-color coded processed image, in which F-actin is also seen in apical regions after MAP4K4 inhibition (Fig. 2k), and (3) those perpendicular F-actin are enriched in pMLC2, indicating that they are highly contractile (Fig. 2j). We think that this data confirms that actomyosin contractility increases tensile stresses at cell-cell junction in MAP4K4 LOF.

5. Some answers to the last point are tentatively offered in the last set of experiments in figure 6. Here, MAP4K4 is shown to be transiently and dynamically recruited to the cell-cell junction. It is not clear, however, what MAP4K4 does there. The authors propose that MAP4K4 promotes junctional turnover and, consistent with this idea, they show that the ectopic expression of the proteins enhances cell scattering. However, it would seem necessary to directly study the impact of MAP4K on junctional stability by directly measuring, for example, the turnover of junctional proteins (such as E-cadherin) through, FRAP experiments. Some clues as to the molecular mechanisms through which MAP4K4 operates at the cell-cell junction would also seem required (see general comments).

We agree with the reviewer that monitoring the turnover of E-cadherin is important. As suggested, we performed FRAP analysis of E-cadherin on MAP4K4 LOF and overexpression conditions (Extended Data Fig. 6d and e). As cell-cell junctions are stable in MAP4K4 LOF, we anticipated that the turnover of E-cadherin should not change. That is indeed what we observed. However, we anticipated that MAP4K4 overexpression should lead to an increase in E-cadherin turnover as junctions are destabilized. Unexpectedly, we haven't found differences in the recovery rates after MAP4K4 overexpression. However, it has to be noticed that we were not able to perform FRAP experiments in the cells expressing high levels of MAP4K4 as the cell-cell junctions are too unstable in these conditions. We add this paragraph in the discussion:

"It was reported that tension loading can either stabilize or reduced E-cadherin dynamics at cell-cell junction, depending on the model and the mechanism analyzed. Despite the predicted link between adhesion stability and cadherin turnover rates, we could not observe significant difference on cadherin recovery by FRAP analysis in MAP4K4 LOF cells. We noticed that high levels of MAP4K4 expression promotes very dynamic cell-cell junction, making it technically challenging to photobleach, while lower expression levels presented no difference in recovery levels when compared to control. Therefore, we believe that MAP4K4 effect on adhesion stability or Cadherin turnover rates depend largely to its level of expression, and the ideal condition to evaluate Cadherin turnover is challenging to capture. Moreover, for a more comprehensive understanding of the molecular mechanism regulated by MAP4K4 at cell-cell adhesions, the recovery rates of several proteins in the junctional complex should be analyzed, since their dynamics may vary independently, specially under different tension loading and actin binding status."

6. Of note, the author, in Figure 6e, used a structure-function reconstitution approach to investigate the impact of critical

domains of MP4K4 on junctional morphology. It would seem necessary to complement this experiment with an analysis of cluster dynamics and migration.

See comment 2.

1. Huveneers, S., et al., *Vinculin associates with endothelial VE-cadherin junctions to control force-dependent remodeling*. J Cell Biol, 2012. **196**(5): p. 641-52.
2. Andriopoulou, P., et al., *Histamine induces tyrosine phosphorylation of endothelial cell-to-cell adherens junctions*. Arterioscler Thromb Vasc Biol, 1999. **19**(10): p. 2286-97.
3. Dejana, E., F. Orsenigo, and M.G. Lampugnani, *The role of adherens junctions and VE-cadherin in the control of vascular permeability*. J Cell Sci, 2008. **121**(Pt 13): p. 2115-22.
4. Plutoni, C., et al., *Misshapen coordinates protrusion restriction and actomyosin contractility during collective cell migration*. Nat Commun, 2019. **10**(1): p. 3940.
5. Lewellyn, L., M. Cetera, and S. Horne-Badovinac, *Misshapen decreases integrin levels to promote epithelial motility and planar polarity in Drosophila*. J Cell Biol, 2013. **200**(6): p. 721-9.
6. Ndubaku, C.O., et al., *Structure-Based Design of GNE-495, a Potent and Selective MAP4K4 Inhibitor with Efficacy in Retinal Angiogenesis*. ACS Med Chem Lett, 2015. **6**(8): p. 913-8.
7. Vitorino, P., et al., *MAP4K4 regulates integrin-FERM binding to control endothelial cell motility*. Nature, 2015. **519**(7544): p. 425-30.
8. Nam, G.S., et al., *A new function for MAP4K4 inhibitors during platelet aggregation and platelet-mediated clot retraction*. Biochem Pharmacol, 2021. **188**: p. 114519.
9. Fiedler, L.R., et al., *MAP4K4 Inhibition Promotes Survival of Human Stem Cell-Derived Cardiomyocytes and Reduces Infarct Size In Vivo*. Cell Stem Cell, 2020. **26**(3): p. 458.
10. Ammirati, M., et al., *Discovery of an in Vivo Tool to Establish Proof-of-Concept for MAP4K4-Based Antidiabetic Treatment*. ACS Med Chem Lett, 2015. **6**(11): p. 1128-33.
11. Oakes, P.W., et al., *Tension is required but not sufficient for focal adhesion maturation without a stress fiber template*. J Cell Biol, 2012. **196**(3): p. 363-74.
12. Tripolitsioti, D., et al., *MAP4K4 controlled integrin beta1 activation and c-Met endocytosis are associated with invasive behavior of medulloblastoma cells*. Oncotarget, 2018. **9**(33): p. 23220-23236.
13. Yue, J., et al., *Microtubules regulate focal adhesion dynamics through MAP4K4*. Dev Cell, 2014. **31**(5): p. 572-85.
14. Roberto, G.M. and G. Emery, *Directing with restraint: Mechanisms of protrusion restriction in collective cell migrations*. Semin Cell Dev Biol, 2022. **129**: p. 75-81.

June 6, 2023

RE: Life Science Alliance Manuscript #LSA-2023-02196-T

Mr. Gregory Emery
Institute for Research in Immunology and Cancer
C.P. 6128, succursale Centre-ville
Montreal, Quebec H3C 3J7
Canada

Dear Dr. Emery,

Thank you for submitting your revised manuscript entitled "MAP4K4 regulates forces at cell-cell and cell-matrix adhesions to promote collective cell migration". We would be happy to publish your paper in Life Science Alliance pending final revisions necessary to meet our formatting guidelines.

- please add the Twitter handle of your host institute/organization as well as your own or/and one of the authors in our system
- please ensure that the Authors' names in the system and manuscript file match (Anna Clouvel-Gervaiseau in the system vs. Anna Clouvel in the manuscript file)
- please add an Author Contributions section to your main manuscript text and in the system
- please add a conflict of interest statement to your main manuscript text
- please move your main, supplementary figure, and video legends after the references section
- LSA allows supplementary figures, but no EV Figures; please update your callouts for the Supplementary Figures in the manuscript Fig EV1A=Fig S1A;
- please make sure the manuscript sections are aligned in accordance with LSA's formatting guidelines: please separate the Figure legends and Supplemental Figure legends into separate sections
- please name the sections 'Figure Legends' and 'Supplementary Figure Legends'
- you may consider uploading Figure 7 as a Graphical Abstract rather than as a figure, but this is up to you

Figure checks:

- please indicate the molecular weight next to each protein blot

A. FINAL FILES:

B. MANUSCRIPT ORGANIZATION AND FORMATTING:

Sincerely,

June 16, 2023

RE: Life Science Alliance Manuscript #LSA-2023-02196-TR

Mr. Gregory Emery
Institute for Research in Immunology and Cancer
C.P. 6128, succursale Centre-ville
Montreal, Quebec H3C 3J7
Canada

Dear Dr. Emery,

Thank you for submitting your Research Article entitled "MAP4K4 regulates forces at cell-cell and cell-matrix adhesions to promote collective cell migration". It is a pleasure to let you know that your manuscript is now accepted for publication in Life Science Alliance. Congratulations on this interesting work.

DISTRIBUTION OF MATERIALS:

Again, congratulations on a very nice paper. I hope you found the review process to be constructive and are pleased with how the manuscript was handled editorially. We look forward to future exciting submissions from your lab.

Sincerely,
